# DeepLabStream enables closed-loop behavioral experiments using deep learning-based markerless, real-time posture detection

Jens F. Schweihoff [1], Matvey Loshakov[1], Irina Pavlova[1], Laura Kück[2], Laura A. Ewell[2] & Martin K. Schwarz [1✉]

In general, animal behavior can be described as the neuronal-driven sequence of reoccurring postures through time. Most of the available current technologies focus on offline pose estimation with high spatiotemporal resolution. However, to correlate behavior with neuronal activity it is often necessary to detect and react online to behavioral expressions. Here we present DeepLabStream, a versatile closed-loop tool providing real-time pose estimation to deliver posture dependent stimulations. DeepLabStream has a temporal resolution in the millisecond range, can utilize different input, as well as output devices and can be tailored to multiple experimental designs. We employ DeepLabStream to semi-autonomously run a second-order olfactory conditioning task with freely moving mice and optogenetically label neuronal ensembles active during specific head directions.

[1] Functional Neuroconnectomics Group, Institute of Experimental Epileptology and Cognition Research, Medical Faculty, University of Bonn, Bonn, Germany.
[2] Institute of Experimental Epileptology and Cognition Research, Medical Faculty, University of Bonn, Bonn, Germany. ✉email: Martin.Schwarz@ukbonn.de

A major goal in behavioral neuroscience is the correlation of behavioral expressions with neuronal activity. For best effectiveness, however, the behavior should be identified in real-time, allowing for instantaneous feedback, i.e., closed-loop manipulation based on the current behavioral expression[1,2]. Currently, such experimental systems often rely on specialized, on-purpose setups, including intricate beam brake designs, treadmills, and virtual reality setups to approximate the movement of the investigated animal in a given environment and then react accordingly[3–11].

Classic manipulations of neuronal activity such as lesions, transgenic alterations, and pharmacological injections result in long-lasting, and sometimes chronic changes in the tested animals, which can make it difficult to interpret behavioral effects. In recent years, there has been a shift towards techniques that allow for fast, short-lived manipulation of neuronal activity. Optogenetic manipulation, for example, offers high temporal precision, enabling the manipulation of experience during experimental tasks that test mechanisms of learning and memory[12–14], perception[15,16], and motor control[17,18]. Such techniques offer a temporal resolution precise enough that the neuronal manipulation can match the timescale of either behavioral expression or neuronal computation.

Recent developments in the field of behavioral research have made offline pose estimation of several species possible using robust deep learning-based markerless tracking[19–21]. DeepLabCut (DLC)[19], for example, uses trained deep neural networks to track the position of user-defined body parts and provides motion tracking of freely moving animals. Additionally, sophisticated computational approaches have allowed for disentangling the complex behavioral expressions of animals into patterns of reoccurring modules[22–26]. In vivo single-unit recording[27], along with recent advances in in vivo voltage imaging[28] and miniaturized calcium imaging techniques[29–31], facilitate real-time measurements of neuronal activity in freely moving mice. Together, these techniques provide a platform for correlating recorded neuronal activity and complex behavior.

We here introduce DeepLabStream (DLStream), a multipurpose software solution that enables markerless, real-time tracking, and neuronal manipulation of freely moving animals during ongoing experiments. Its core capability is the orchestration of closed-loop experimental protocols that streamline posture-dependent feedback to several input, as well as output devices. We modified state-of-the-art pose estimation based on DLC[19] to be able to track the postures of mice in real-time. To demonstrate the software's capabilities, we conducted a classic, multilayered, freely moving conditioning task, as well as a head direction-dependent optogenetic stimulation experiment using a neuronal activity-dependent, light-induced labeling system (Cal-Light)[1]. Finally, we discuss the versatility of DLStream to adapt to different experimental conditions and hardware configurations.

## Results

DLStream enables closed-loop stimulations directly dependent on actual expressed behavioral postures. Our solution is fully autonomous and requires no additional tracking-, trigger- or timing-devices. All experiments can be conducted without restriction to the animal's movement and each experimental session is run fully autonomously after the first setup. Initially, we trained DLC-based pose estimation networks offline for each experimental environment and then integrated them into DLStream (see "Methods" section). Briefly, frames were taken from a video camera stream, and analyzed using an integrated deep neural network, trained using the DLC framework. Next, the

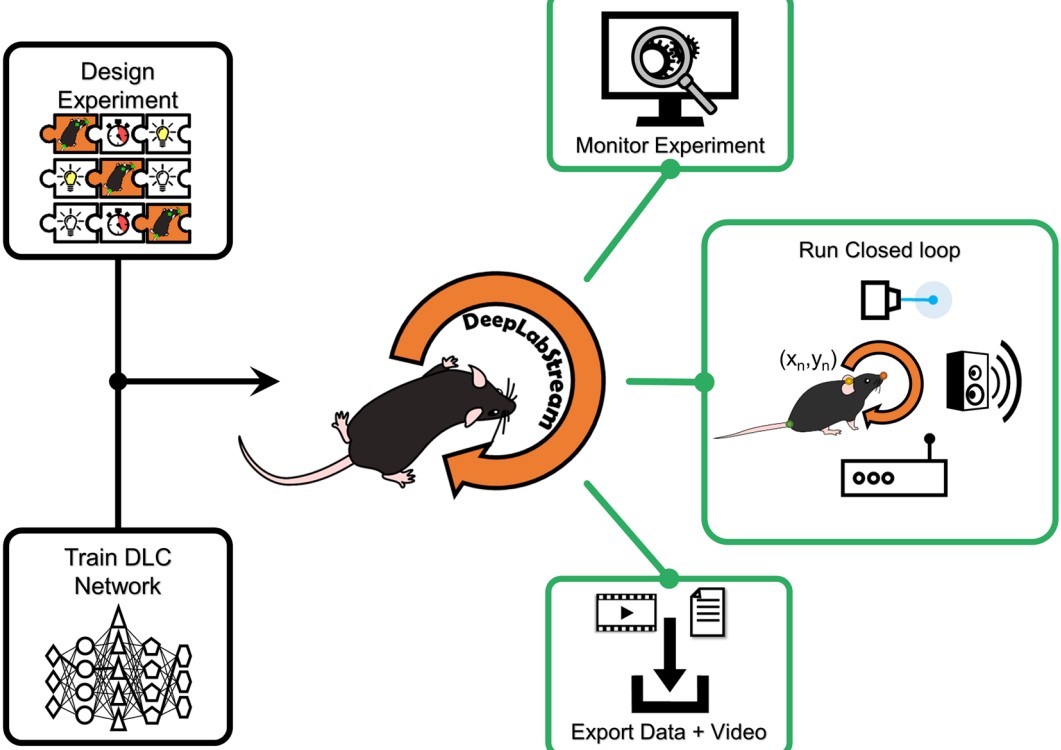

**Fig. 1 A visual representation of DLStream.** Visual representation of workflow in DLStream. Initially, an experimental protocol is designed using a sequence of modules (puzzle pieces) and a trained DLC network is integrated into DLStream. Afterward, DLStream provides three different outputs for every experiment. 1. Experiments can be monitored on a live stream. 2. The experimental protocol is run based on posture detection 3. Recorded video and experimental data are exported after the experiment is done.

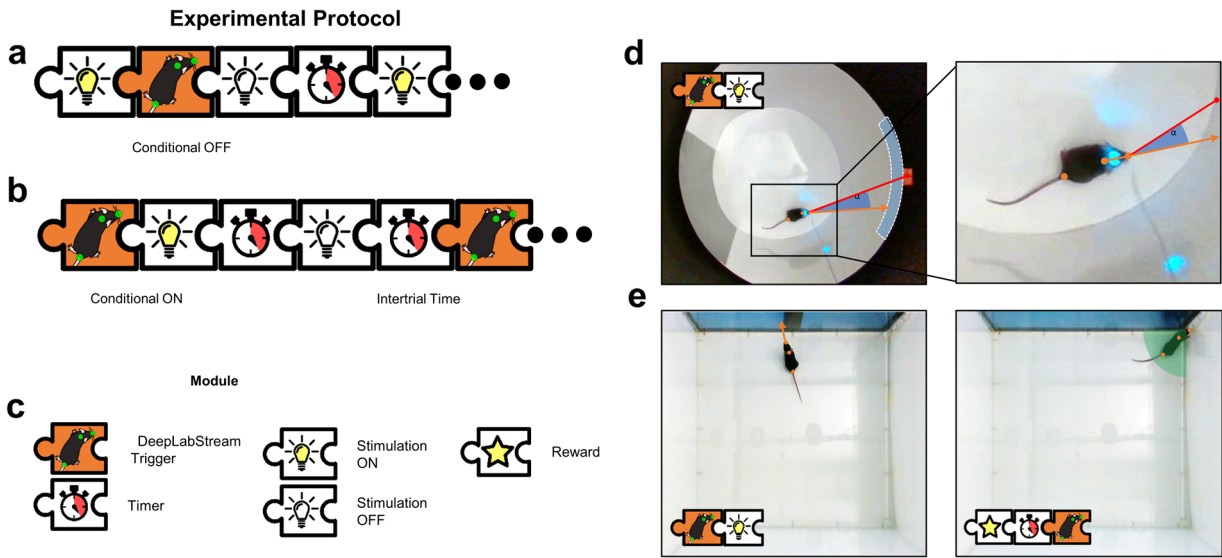

**Fig. 2 Experimental design using DLStream. a**, **b** Schematic design of an experimental protocol with a posture-based trigger. Manipulation can be turned "Conditional OFF" (**a**) and "Conditional ON" (**b**) based on the mouse's behavior. The combination of several modules allows building a sophisticated experimental protocol. For example, the timer module can be utilized to design inter-trial and -stimulus timers (**b**), minimum stimulation (**b**), or delayed triggers (**e**). **c** Description of available modules in **a** and **b**. **d** Application of the above-described design in an optogenetic experiment. The stimulation is triggered dependent on head direction angle (orange arrow, α) to a reference point (red line) within the target window (blue arc). **e** Application of the above-described design in a classical conditioning task. The mouse is shown an image when looking at the screen (left) and the reward is removed if it does not move into the reward location within a predefined timeframe (right, green zone). The mouse's posture is shown with orange dots.

resulting pose estimation was converted into postures and transferred to an additional process that supervises the ongoing experiment and outputs feedback to connected devices (Fig. 1). As a result, experiments run by DLStream comprise a sequence of modules (Fig. 2c) depending on the underlying experimental protocol. Basic modules, such as timers and stimulations, are posture-independent and control fundamental aspects of the experiment. Timers keep track of the time passing during frame-by-frame analysis and act as a gate for posture-based triggers and stimulations (e.g., inter-stimulus time). Stimulations specify which devices are triggered and how each device is controlled once it was triggered (e.g., reward delivery). Posture-based triggers are sets of defined postures (e.g., position, head direction, etc.) that initialize a predefined cascade (stimulation) once detected within an experiment (see Fig. 2 for examples). As an experiment is conducted, DLStream records and subsequently exports all relevant information, including posture tracking, experimental status, and response latency in a table-based file. During any experiment, posture tracking is visualized on a live video stream directly enabling the monitoring of the conducted experiment and tracking quality. Additionally, the raw video camera stream is timestamped and recorded, allowing high-framerate recording, with lower-framerate closed-loop posture detection to save processing power (Fig. 1).

**Classical second-order conditioning using DLStream**. To comprehensively test DLStream we first designed a semi-automated classical second-order conditioning task (Fig. 3a–e). Using DLStream, mice were trained to associate two unknown odors (rose and vanillin) with two visual stimuli, which were initially associated with either a reward or an aversive tone (Fig. 3a). We subsequently tested the conditioned mice in an odor preference task. In the first conditioning stage, DLStream triggered trials when a mouse was facing the screen. For this, a trigger module was designed that utilizes the general head direction of mice, activating stimulation modules only when

mice were looking towards the screen in a 180° window. The mice were conditioned to associate two unknown visual stimuli (a high-contrast black and white image) with a reward or an aversive tone (Fig. 3a) using combinations of predefined stimulation modules. In the positive trial, DLStream delivered a liquid reward by triggering the corresponding stimulation module in a fixed reward location and withdrew it if it was not collected within a preset time period monitored with a timer module. In the negative trial, DLStream delivered only the aversive tone (Fig. 3a). All mice ($n = 10$) were trained for 13 days and selected based on their individual performance to reach the success criterion (85% reward collection within one session, $n = 6$ mice). We limited the number of sessions to 1 h or 40 trials per day within our experimental protocol. Note that no mouse needed more than 45 min to complete a session. During the subsequent second-order conditioning, the mice were presented with two novel odors (rose and vanillin), placed in a petri dish in front of the screen (Fig. 3b). Visual stimuli were previously paired with an odor and pairing was kept throughout all experiments. Upon exploration of one of the two presented odors, DLStream showed the mice the paired, previously conditioned visual stimulus (Fig. 3b). The session was completed when DLStream detected that the mice had explored both odors at least 10 times, or after 10 min had passed. Second-order conditioning was then conducted in two stages. The first stage consisted of the mouse being in direct contact with the odor location (petri dish), while the second was dependent on the proximity of the mouse to one of the locations and its head direction (Fig. 3b). For this, trigger modules designed to detect proximity and the heading direction of mice were used. Each stage was repeated twice with exchanged odor locations.

We then tested for successful second-order conditioning by conducting an offline odor preference task (Fig. 3c). Mice ($n = 6$) were placed in an open field arena with one odor in each of the quarters. In addition to the two conditioned odors, two novel odors (acetophenone and valeric acid) were presented. Mice were given 10 min twice to explore and total investigation time was

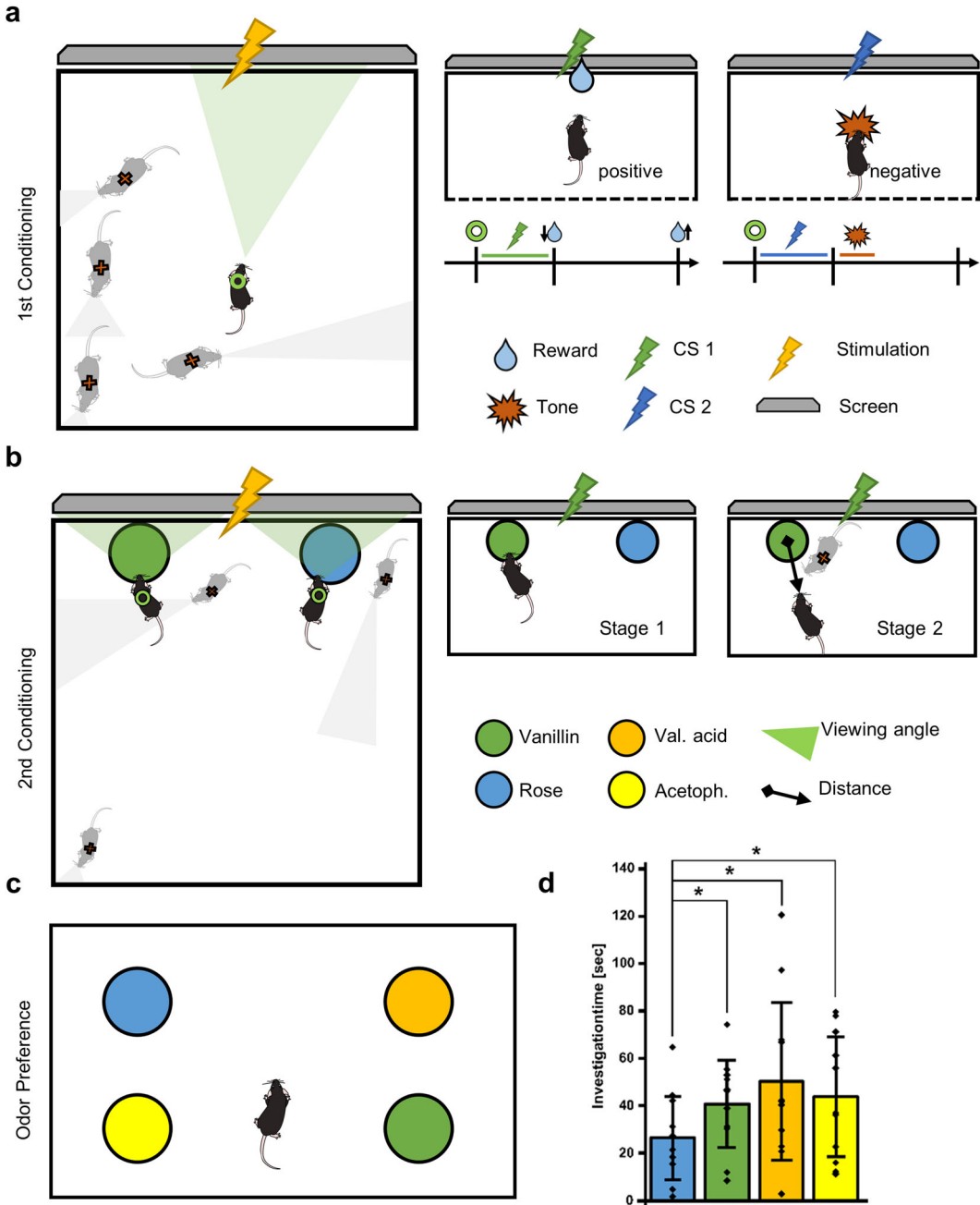

**Fig. 3 Closed-loop conditioning task. a** Conditioning. When a trial is triggered by the mouse facing the screen (green triangle and ring), the mouse is shown a visual stimulus (yellow lightning bolt). Mice not facing the screen do not receive the stimulus (red x). In the positive trial (green lightning bolt, green line), a reward is delivered (blue drop, arrow down) and withdrawn (blue drop, arrow up) if not collected within a preset time period. In the negative trial (blue lightning bolt, blue line) only a loud tone (red polygon) is delivered. **b** 2nd Order conditioning. Upon exploration of either odor location (colored black circle) the mouse is shown one of the previously conditioned visual stimuli on the screen (yellow lightning bolt). Conditioning was conducted in two stages. The first stage (Stage 1) consisted of direct contact with the odor location, while the second (Stage 2) was dependent on the proximity of the mouse to one of the locations (black arrow) and the mouse facing towards it. **c** Odor preference task. The mouse was set in an open field arena with one odor in each of the quarters (colored circles). The total investigation time of each odor source was measured. **d** Investigation time during odor preference task in odor location: ROIs encircling the odor location. The bar graph shows the STD and individual data points. $p < 0.05$ (*) one-tailed paired $t$-test; R/V $p = 0.0395$, R/VA $p = 0.0497$, R/A $p = 0.0311$; $n = 11$ trials (2 trials per mouse, 1 trial excluded, 6 mice total; see also Supplementary Data 1). Error bars represent standard deviation. V = Vanillin (S+), R = Rose (S−), VA = Valeric acid, A = Acetophenone.

measured by taking the circular odor location into account. Mice showed a preference towards the positively conditioned (S+) odor compared to the negatively conditioned (S−) odor by spending significantly more time at the S+ odor location than in the S− odor location (Fig. 3d). While the investigation time of the S− odor was significantly less compared to the investigation time

of both novel odors, we found no significant difference between the S+ odor and the novel odors.

**Optogenetic, head direction-dependent labeling of neurons using DLStream.** As a second example of DLStream's

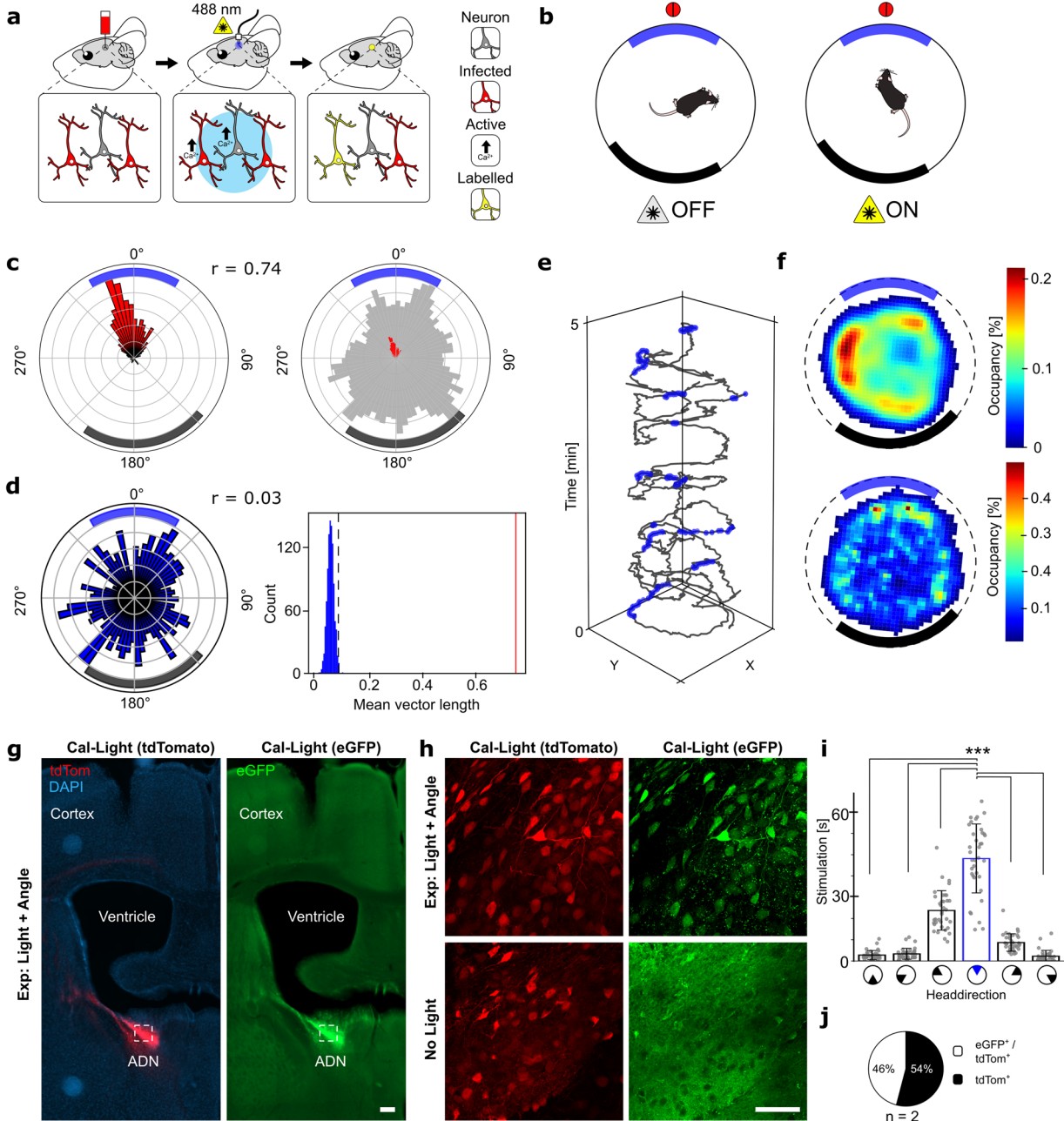

applicability, we tested the possibility to optogenetically label active neurons in the anterior dorsal nucleus of the thalamus (ADN) dependent on the mouse's head direction using the neuronal activity-dependent labeling system Cal-Light[1]. Activity within ADN neurons is known to be modulated by the angle of head direction[27]. Within a stable environment, the angular tuning curve of an ADN neuron remains constant, facilitating experimental designs that span several days[32]. To label ADN ensembles, we utilized DLStream to deliver light stimuli within precisely defined head direction angles (target window) (Fig. 4). The timing was controlled by designated timer modules controlling the onset and offset of light stimulation once the stimulation module was triggered. Mice were placed in a circular white arena with a single black cue at one side and allowed to investigate the arena in one 30-min session per day for four consecutive days. During each session, mice were stimulated via a chronically implanted optical fiber with blue light (488 nm) triggered by their head direction angle. Mice were able to freely move their heads in all directions,

but stimulation was limited to periods when they oriented their head to the designated head direction target window (60° to reference point, Fig. 4b, c and Supplementary Fig. 4). Each stimulation lasted 1–5 s depending on the time spent orienting to the target window (60°) with a minimum inter-stimulus time of 15 s. In the case of the inter-stimulus timer, the module blocked the link between the trigger module and the stimulation module when activated, disabling posture-dependent stimulation for its designated duration.

The resulting average light stimulation per session (48 ± 10 s) occurred selectively in the target angle window across all experimental animals (Fig. 4h). Note that stimulation with outside-target head direction angles can result from individual stimulations having a chosen minimum duration of 1 s, in which the mouse theoretically could sweep its head away from the target window. The average total stimulation time across all four sessions was 357 ± 53 s ($n = 10$ mice). As a control, a yoked group of mice was run such that each mouse regardless of its actual head

**Fig. 4 Optogenetic labeling of head direction-dependent neuronal activity. a** Left: Stereotactic delivery of Cal-Light viruses into the ADN and fiber ferrule placement. Middle: Infected neurons (red) are stimulated with blue light (488 nm) controlled by DLStream. Right: Infected neurons are only labeled (yellow) when they are active (black arrow) during light stimulation (middle). **b** Schematic drawing of the circular arena with the visual cue (thick black arc) and the target window (thick blue arc) around the reference point (red circle). DLStream triggered stimulation is strictly dependent on the correct head direction (blue arc). **c** Left: Representative example (see also Supplementary Data 2) radial histogram of all head directions during stimulation (red) within one session (normalized to the maximum value). The mean resultant vector length is indicated by r. Right: Radial histogram of all head directions during the whole session (gray) and during stimulation (red) (normalized to the maximum value of the entire session). Rings represent quantiles in 20% steps. **d** Left: Representative random sample of the whole session simulating stimulation without DLStream control at random time points during the session (normalized to the maximum value). The mean resultant vector length is indicated by r. For each session, random distributions were calculated 1000 times. Right: For one session, the distribution of mean resultant vector lengths generated by random sampling ($n = 1000$). The red line denotes the actual mean resultant vector length during stimulation in the session. The dotted black line represents the $p < 0.01$ cutoff. **e** Representative example of the mouse's position (gray) over time during the first 5 min of the session in **c**. The stimulation events are shown in blue. **f** Heatmaps representing the relative occupancy of the mouse within the arena during the whole session (top) and stimulation (bottom) in **c**. Cue and target window are shown in their relative position. **g** Example of Cal-Light expression in an experimental mouse. Left: tdTomato expression (red) indicating expression of Cal-Light viruses with nucleus staining (DAPI, blue). Right: Activity-dependent and light-induced eGFP expression (green). The white box represents the zoomed-in region in **h**. The bar represents 200 μm. **h** Close up from **g** vs. a similar region in an animal that was not stimulated with light (no light). Left: tdTomato expression (red). Right: Activity dependent and light-induced eGFP expression (green). The bar represents 50 μm. Note that control mice show no eGFP expression. **i** Average light stimulation during each session (40 total) corresponding to head direction (60° bins) with target window (blue) indicating the DLStream triggered stimulation onset (see also Supplementary Data 3). Paired student's t-test: $p < 0.001$. $n = 10$ mice. Error bars represent standard deviation. **j** Ratio between infected neurons (tdTom$^+$) and activity-dependent labeled neurons (eGFP$^+$/tdTom$^+$) in mice matching selection criteria (see "Methods" section). $n = 2$ mice.

direction, received the exact same temporal stimulus as a paired experimental mouse. Therefore, in the yoked group, light stimuli were decoupled from head direction (Supplementary Fig. 1a).

Mice explored the entire arena during the task and the resulting light stimulation was not dependent on the animal's position in the arena, as animals could angle their head in the target orientation from any position within the arena (Fig. 4e, f). Randomly sampling angles equal to the number of stimulated angles, revealed a nonspecific distribution of angles—i.e., mice oriented in all directions (Fig. 4d, left). Note that for each individual mouse, the mean resultant length for stimulated angles was significantly larger than would be expected by random sampling (see "Methods" section, $n = 1000$ samples, $p < 0.01$) (Fig. 4d, right).

Next, we quantified the percentage of ADN neurons that were labeled in three different groups (experimental, no light, and yoked). Only mice that matched selection criteria (correct fiber ferrule placement as well as injection placement) were taken into account when quantifying Cal-Light conversion (see "Methods" section and Supplementary Fig. 2 for details). Cal-Light infected neurons showed a 46% conversion within the ADN (Fig. 4j, $n = 2$ mice) while mice receiving no light stimulation but underwent the same sessions had no light-induced labeling present (Fig. 4g–j). Furthermore, within the yoked group, only a very low percentage (~4%, $n = 2$ mice) labeling was observed (Supplementary Fig. 1b, c), indicating that the repeated pairing between light stimulation and head direction triggered activity was essential for Cal-Light-mediated fluorescent labeling.

**Computational performance of DLStream**. A reality of any closed-loop system is that there are temporal delays between real-time detection of particular postures and stimulus output. To address this challenge, we first rigorously defined the variance of behavioral parameters we are measuring. To estimate the spatiotemporal resolution of postures that can be detected using our integrated network configuration, we compared the pose estimation error of our networks and the correlated parameter changes between frames. Note that, due to the inherent individual network performances, DLStream's effective accuracy in posture detection is heavily influenced by the previous training of utilized networks. Nevertheless, if performance is not sufficient for the executed experiment, DLC networks can always be retrained

using the DLC provided tools. In our hands, the trained network used during optogenetic experiments resulted in an estimated average pose estimation error of $4 \pm 12$ pixels (px) for the neck point, $3.3 \pm 4.4$ px for the nose, and $3.3 \pm 2.0$ px for tail root ($n = 597$ images) when compared to a human annotator labeling the same data set (mice without tail were ~60 px long in our 848 × 480 px recordings). Body part estimation resulted in an average head direction variance of $3.6 \pm 9.6°$ (tested in 80 sessions for 1000 frames per session) between consecutive frames with an estimated average error of $7.7 \pm 15.1°$ compared to human annotation ($n = 597$, ground truth) per frame. The frame-by-frame variance is a product of performance errors and the inhomogeneous movement of the animal during experiments while the difference between network pose estimation and human annotation is most likely a result of inaccurate tracking which can be reduced by additional training and/or bigger training sets. Note that depending on the mixture of episodes of fast movements and slow movements during sessions, the variance might change. We next manually evaluated posture detection accuracy during optogenetic experiments and found a false-positive rate of 11.8%. In the evaluated sessions most, false-positive events were anomalies in mouse behavior such as spontaneous jumping, that can possibly be further reduced by additional network training if necessary. Additionally, we estimated the general false-positive/false-negative rate for our head direction trigger based on a human-labeled data set and found a false negative rate of $11.1 \pm 4.1\%$, while false-positive rates were $11.6 \pm 4.8\%$ ($n = 597$; see Supplementary Fig. 3 for additional data).

During optogenetic experiments ($n = 80$), DLStream reached an average performance time of $33.32 \pm 0.19$ ms per frame, matching the average camera framerate of 30 Hz (33.33 ms), including posture detection and computation of the resulting experimental protocols until output generation. We also measured the hardware latency to estimate the time between posture detection and triggered stimulation during optogenetic sessions from three different mice ($n = 164$ stimulation events). Here, the resulting light stimulation occurred within 5 frames ($4.8 \pm 1.1$ frames at 30 fps; ≈150 ms). It is important to consider here that the total latency critically depends on the individual setups and the intrinsic parameters of connected components. To evaluate the limits of DLStream, we tested different hardware configurations and investigated performance levels and response time.

First, average performance was measured during 10,000 frames in two different configurations with two different camera settings (30 fps and 60 fps with 848 × 480 px resolution) using the same camera used in our experiments. With the standard 30 fps camera setting, the advanced configuration (Intel Core i7-9700K @ 3.60 GHz, 64 GB DDR4 RAM, and NVidia GeForce RTX 2080 Ti (12GB) GPU) achieved reliable 30 fps (33.33 ms per frame) real-time tracking with 30 ± 7 ms, while the other system (Intel Core i7-7700K CPU @ 4.20 GHz, 32GB DDR4 RAM and NVidia GeForce GTX 1050 (2GB) GPU) only reached an average analysis time of 91 ± 10 ms. Using a higher framerate input from the camera (60 fps; 16.66 ms per frame), the overall performance did not change considerably (24 ± 9 and 90 ± 9 ms, respectively). Second, we tested a different camera (Basler acA1300 – 200 µm), which lacks the depth capabilities of the Intel RealSense camera but comes with increased framerate, on the advanced configuration with different image resolutions (ranging from 1280 × 1014 to 320 × 256 px) to benchmark DLStream's upper-performance limits with more standardized cameras and resolutions. While we initially used DLC trained ResNet50[33,34] networks during experiments, we additionally evaluated the capabilities of the other available models (ResNet101[33,34], MobileNetv2[35]) and also a higher number of body parts (3, 9, and 13 body parts). In our hands, DLStream's latency reached a maximum of 130 ± 6 Hz (ca. 8 ms) with the MobileNetv2 architecture at 320 × 256 px resolution, while the ResNet50 network reached its upper limit at 94 ± 6 fps (ca. 10 ms) at the same resolution (see Supplementary Table 1 for more details).

## Discussion

There has been a recent revolution in markerless pose estimation using deep neural networks. However, these system's intrinsic design delays analysis until after the end of the experiment owing to their heavy computation. Here we take advantage of the power of DLC's offline body part tracking to train a neural network and integrate it into our real-time, closed-loop solution.

As observers, experimenters often record and interpret an animal's behavior by taking its movement as an approximation of the underlying intention or state of mind. Building on this generalization, behavior can be defined, categorized, and even sequenced by examining estimations of the animal's movement[23,24,36,37]. Classified periods of behavior, so-called behavior modules, are commonly used for offline quantification (e.g., phenotyping). In addition, behavior modules are also very promising in closed-loop approaches to react specifically to complex behavior. Such an analysis yields the prospect of predicting behavior, for example by matching initial elements of a uniquely arranged behavioral sequence. With DLStream, a combination of triggers based on the animal's posture or a sequence of postures can be integrated into experimental designs. Example triggers include center-of-mass position, direction, and speed of an animal, although multiple individual tracking points can also be utilized, such as the position and trajectory of multiple, user-defined body parts. This allows the design of advanced triggers that include head direction, kinematic parameters, and even specific behavior motifs (e.g., rearing, grooming, or sniffing). Out of the box, DLStream supports triggers based on single-frame as well as sequential postural information, although complex behavior modules could also be utilized once behavior based on collected posture data has been classified, modeled, and integrated as custom trigger modules into DLStream. The challenge in manually designing triggers for relevant behavior is similar to the challenges faced in offline analysis, where it has already been done for a variety of relevant read-outs, such as described in VAME[25], B-SOID[26], and SIMBA[38]. While this is relatively simple when only single-frame posture detection with low-level features are utilized (e.g., head direction angle), defining sequential changes of features to capture more complex changes in the animal's movement requires the careful exploration and extraction of relevant features. Once feature extraction is established, however, the behavior of interest can be detected and implemented as a custom trigger into DLStream. Promising approaches for machine-guided classification are being actively developed using DLC-based pose estimation as input[25,26,38], which should increase the range of available triggers considerably. The integration of fast behavior classifiers, for example, would enable the design of a trigger that reacts to complex behaviors without the need for a strict, manual description of relevant feature changes. To facilitate the design of custom experiments and triggers, we offer several tutorials and guides with our DLStream code (https://github.com/SchwarzNeuroconLab/DeepLabStream). Additionally, easy-to-use, GUI-based toolkits such as SimBA[38] facilitate the generation and open-source distribution of robust classification models.

Two of the most considerable limitations in all real-time applications are the latency of the system to react to a given input and the rate at which meaningful data are obtained. While the latency is dependent on the computational complexity, the rate is dependent on several factors, and hardware constraints in particular. A researcher might only need the broadest movements or behavioral states to understand an animal's basic behavior, or fast, accurate posture sequences to classify behavioral modules on a sub-second scale[24,36]. Considering that animals behave in a highly complex manner, a freely moving approach is favorable since restricting movement likely reduces the read-out of the observable behavioral spectrum.

DLStream is designed as a universal solution for freely moving applications and can, therefore, be used to investigate a wide range of organisms. DLC networks already have the innate capability to track a variety of animals across different species[39] which can be directly translated to experiments within DLStream. Additionally, its architecture was designed for short to mid-length experiments (minutes to hours). There are no built-in limitations to conduct long-lasting experiments (days to weeks), but DLStream currently lacks the capability to automatically process the large amounts of raw video data or other utilities that become necessary when recording for longer periods of time. One possible solution would be to remove the raw video output and only save the experimental data that includes posture information, which would considerably lighten the necessary data storage space.

With regards to latency, the current fully tested, closed-loop timescale enables the tracking and manipulation of a wide range of activities a rodent might perform during a task. Very fast movements, however, like whisker movement[40,41] and pupil contraction[42,43] might not be fully detected using the 30 Hz configuration from our experiments, but might be possible using lower camera resolution and a different network architecture (e.g., MobileNetv2; Supplementary Table 1). Most freely moving applications usually lack the resolution to visualize whiskers and pupils while maintaining an overview of the animal's movement in a large arena. Note that offline analysis of raw, higher frame-rate videos can still be recorded if desired. DLStream is able to take frames from a higher framerate stream but still maintain a lower, loss-less closed-loop processing rate. On a side note, developments in alternative, non-video-based, specialized tracking (e.g., eye-tracking[44]) might lead to a solution for researchers interested in capturing truly holistic behavioral data.

Using posture-dependent conditioning, mice were able to successfully learn an association between a visual stimulus and a reward, thus demonstrating DLStream's capabilities with respect to the automatization of classical learning tasks. Second-order

conditioning resulted in an odor preference between the conditioned odors. Importantly, mice did not need any previous training apart from initial habituation to the reward delivery system to perform this task. Interestingly, mice investigated the novel odors at the same level as the positively reinforced odor, suggesting a novelty component that influences the animal's investigative behavior. It is likely that previous habituation to the neutral odors would reduce that effect. Importantly, possible applications are not limited to classical conditioning tasks. Many behavioral tasks, an operant conditioning task, for example, could also be accomplished by setting a specific posture or sequence of postures as a trigger to reward, punish or manipulate freely behaving animals during an experimental session.

To dissect and better understand the neuronal correlates of complex behaviors a better understanding of the actively participating neuronal assembles is desirable. Techniques that can bridge connectomics, electrophysiology, and ethology hold the potential to reveal how computations are realized in the brain and subsequently implemented to form behavioral outcomes. For instance, by utilizing neuronal activity-dependent labeling systems such as Cal-Light[1], Flare[2], or CaMPARI[45], it is already possible to visualize active neurons during episodes of behaviors of interest. However, the identification of repetitive/reoccurring episodes and following activation of a specific trigger is currently restricted by a lack of dynamic closed-loop systems. With DLStream, we show that the real-time detection of specific behaviors in freely moving mice can be combined with neuronal activity-dependent labeling systems (Cal-Light) to investigate the neuronal correlates of behavior. We here delivered light stimuli to the ADN to label neural ensembles active during specific head directions. Within the selected experimental animals, labeling of active neurons was successful and resulted in the labeling of a subset of cells (ca. 46%). Our goal was to demonstrate that DLStream can potentially label such specific ensembles of active neurons during relevant behavioral expressions. Direct optogenetic activation and inhibition[46–49] of neuronal population based on posture detection might also be possible with DLStream, although our stimulation setup had delays of ~150 ms between detection and manipulation, which may be too slow for certain applications. In our hands, delays were still short enough to allow for targeting activity triggered calcium dynamics by the Cal-Light system[1,50]. Using a solution like DLStream the range of detectable behaviors would increase substantially and applications for action- and posture-dependent labeling and subsequent manipulation of different freely moving species are wide-ranging. Additionally, optimizing the setup might allow faster feedback times as our hardware limited the effective use of the underlying software performance of DLStream.

Comparative tests between our available computer configurations suggest that the GPU power is responsible for major performance gains in real-time tracking utilizing DLStream. CPU power is also important since several parallel processes need to be maintained during complex experimental protocols and processing of pose estimation. DLStream is able to analyze new frames as soon as the current frame is fully processed, therefore a higher framerate does not slow down DLStream but rather enables it to work at the upper-speed limit (Supplementary Table 1). At this stage, the full utilization of higher framerates will heavily depend on the hardware configuration and the experimenter's resolution requirements. From a pure performance perspective, the use of faster neural network architectures (e.g., MobilNetV2[35]) trained within the DLC framework already increases the available framerate by a factor of four (30–130 fps, Supplementary Table 1), which is in line with the recent big-scale benchmark tests run by DLC[51,52] and other publications[11].

DLStream is compatible with old and new versions of DLC. Although originally developed for DLC 1.11 (Nature Neuroscience Version[19]), we have successfully tested the newest DLC version (DLC 2.x[19]) without encountering problems. Networks trained on either version can be fully integrated into DLStream and used as needed. Additionally, DLStream is in principle able to support positional information from other pose estimation networks[20,21] but would currently require some customization by the user as these networks have different input/output formats that would need to be adapted to the current workflow. An experimental implementation of additional pose estimation sources can be found on our GitHub page    (https://github.com/SchwarzNeuroconLab/DeepLabStream). This includes the implementation of models exported by DeepPoseKit[21] (LEAP[20], StackedHourglass[53], StackedDenseNet[21], DLC) and DLC-Live[51] (DLC) as well as multiple animal DLC (maDLC). However, the performance speed of such network implementations needs to be evaluated and compared to established pose estimation speed. Recent developments by DLC regarding online pose estimation reported real-time network performances for architectures used by DLC[51].

With the recent advances in markerless, multiple animal tracking (e.g., maDeepLabCut, SLEAP[20,54], id-tracker[55]) an adaptation of DLStream to include multiple animal-based triggers would further enhance its versatility. In theory, such an adaptation should be similar to using multiple body parts. The challenge will most likely be the precise definition of social triggers and the design of relevant experiments using closed-loop stimulation. We briefly tested closed-loop multiple animal tracking using a pair of differently colored mice (standard DLC), as well as a maDLC-trained network on mice with same-colored fur, and were able to confirm that DLStream can utilize the similar output pose estimation of both. However, full verification of implementation within an experiment is yet to be done.

Notably, DLStream could also be upgraded to use 3D posture detection as for example implemented recently by EthoLoop[10]. To achieve this, two reasonable approaches exist that allow 3D tracking of animals based on video analysis. A DLC native approach would be the use of multiple camera angles to triangulate the animal's position (see ref. [39] for further information). An alternative approach would be the use of depth cameras to estimate the distance of an animal to the camera and thereby generate a 3D representation.

DLStream is a highly versatile, closed-loop software solution for freely moving animals. While we show its applicability in posture-dependent learning tasks and optogenetic stimulation using mice, we see no obvious limitations to the applicability of DLStream on different organisms and other experimental paradigms.

## Methods

**Mice**. C57BL/6 mice were purchased from Charles River (Sulzfeld, Germany) and maintained on a 12-h light/12-h dark cycle with food and water always available. All the experiments were carried out in accordance with the German animal protection law (TierSCHG), FELASA, and were approved by the animal welfare committee of the University of Bonn.

**AAV production**. AAV pseudo-typed vectors (virions containing a 1:1 ratio of AAV1 and AAV2 capsid proteins with AAV2 ITRs) were generated as described[57,58]. Briefly, human embryonic kidney 293 (HEK293) cells were transfected with the AAV cis plasmid, and the helper plasmids by standard calcium phosphate transfection. Forty-eight hours after transfection the cells were harvested and the virus purified using heparin affinity columns (Sigma, St. Louis, MO)[59]. Purification and integrity of the viral capsid proteins (VP1-3) were monitored on a Coomassie-stained SDS/protein gel. The genomic titers were determined using the ABI 7700 real-time PCR cycler (Applied Biosystems) with primers designed to WPRE.

**Surgical procedure**. Viral injections were performed under aseptic conditions in 2-months-old C57BL/6 mice. Mice were initially anesthetized with an oxygen/isoflurane mixture (2–2.5% in 95% O₂), fixed on the stereotactic frame, and kept under a

constant stream of isoflurane (1.5–2% in 95% O₂) to maintain anesthesia. Analgesia (0.05 mg/kg of buprenorphine; Buprenovet, Bayer, Germany) was administered intraperitoneal prior to the surgery, and Xylocaine (AstraZeneca, Germany) was used for local anesthesia. Stereotactic injections and implantations of light fiber ferrules were performed using a stereotactic frame (WPI Benchmark/Kopf) and a microprocessor-controlled minipump (World Precision Instruments, Sarasota, Florida). The viral solution (1:1:2; AAV-TRE-EGFP, Addgene #89875; AAV-M13-TEV-C-P2A-TdTomato, Addgene #92391; AAV-TM-CaM-NES-TEV-N-AsLOV2-TEV-seq-tTA, Addgene plasmid #92392) was injected unilaterally into the ADN. Viruses were produced as previously described. To reduce swelling animals were given Dexamethasone (0.2 mg/kg). For implantation, the skin on the top of the scalp was removed and the skull cleared of soft tissue. Light fiber ferrules were implanted and fixed using a socket of dental cement. Loose skin around the socket was fixed to the socket using tissue glue (3 M Vetbond). Directly after the surgery animals were administered 1 ml 5% Glucosteril solution. To prevent the wound pain, analgesia was administered on the three following days. Animals were left to rest for at least 1 week before starting handling. Experiments were conducted 3 weeks after surgery.

**Perfusion**. Mice were anesthetized with a mixture of Xylazine (10 mg/kg; Bayer Vital, Germany) and ketamine (100 mg/kg; Bela-pharm GmbH & Co. KG, Germany). Using a peristaltic pump (Laborschlauchpumpe PLP33, Mercateo, Germany), the mice were transcardially perfused with 1× PBS followed by 4% paraformaldehyde (PFA) in PBS. Brains were removed from the skull and post-fixed in 4% PFA overnight (ON) at +4 °C. After fixation, the brains were moved into PBS containing 0.01% sodium azide and stored at +4 °C until sectioning. Fixed brains were sectioned coronally (70 or 100 μm) using a vibratome (Leica VT1000 S) and stored in PBS containing 0.01% sodium azide at +4 °C.

**Conditioning task**. Mice were placed in an open field arena (70 × 70 cm). Each session lasted 1 h or a maximum number of 40 trials. A session consisted of a random sequence of trials. Additionally, if an animal successfully finished 20 positive trials, the session was ended. A trial was initiated when the animal was facing the screen. Each trial lasted 20 s with an inter-trial interval of 30 s. At the beginning of each trial, a visual stimulus was shown on the screen for 10 s. In the positive trial, a reward was delivered at the end of the visual stimulus and withdrawn if not collected within 7 s. In the negative trial, a loud tone (100 dB) was delivered and no reward was given. After at least five sessions, animals that learned the association successfully (>85% success rate in the positive trial) were transferred to the next stage. We did not evaluate the success rate of negative trials since the aversive stimulation was delivered regardless of the animal's behavior.

The visual stimulus was a high-contrast, black, and white image of an X or + spanning the whole screen. The screen was the same size as the arena wall it was placed at.

**Second-order conditioning task**. Animals were placed in the open-field arena. Two Petri dishes filled with fresh bedding were placed on the wall facing the screen. Two odorants (10 μl on filter paper) were placed in one of the Petri dishes each. A pair of an odorant and visual stimulus (negative or positive) was chosen and kept throughout the experiments. Upon exploration of an odor location, the animal was shown the corresponding visual stimulus. The session was completed after the animal explored both odors for at least 10 individual times or after 10 min. Conditioning was conducted in two stages. Both stages were repeated with switched odor positions, resulting in a total of four repetitions per animal. The first stage consisted of direct contact with the odor location, while the second was dependent only on the proximity of the animal to a location and the animal facing towards it.

**Preference task**. The mouse was placed in a different open-field arena (70 × 40 cm) with one odor in each of the quarters. In addition to the conditioned odors, two neutral odors were presented. The mouse was given 10 min twice to explore the arena with an inter-trial time of 10 min in between. Total investigation time was measured with circular ROIs, corresponding to the odor location, above each petri dish. Trials in which the mice did not investigate any odor source were excluded (1 trial out of 12; n = 6 mice).

**Head direction-dependent optogenetic stimulation**. Mice were put in a cylindrical white arena with a single cue (a black vertical bar). The arena was enclosed by a black curtain. A random point was chosen to act as a reference for head direction (0°). The reference point was kept constant between experimental sessions and animals but was not visible to the animal. To habituate the animal to the arena, the animal was put into the arena for 30 min for 2 days and reward pellets were placed randomly inside the arena at the 0, 10, and 20 min mark.

*Experimental group*: During the experiment, light stimulation (488 nm, 15 mW; Laser OBIS LX/LS, controlled by OBIS LX/LS Single Laser Remote, Coherent Inc., Santa Clara, CA, USA) was initiated whenever the animal's head direction was within a 60° window around the reference point. Stimulation lasted 1 s or as long as the head direction was maintained in the window up to a maximum of 5 s. After each stimulation, further stimulation was discontinued for at least 15 s to avoid overheating of brain tissue and in line with the originally published Cal-Light experiments[1]. The animal was allowed to investigate the arena over four

consecutive days for 30 min sessions each day during which the animal was stimulated. Animals were perfused 1 day after the last session.

*Yoked group*: In the yoked control group animals were previously paired with another animal from the experimental group. Each control animal received the exact same temporal stimulus as the paired experimental animal, decoupled from its own head direction. Animals were treated and ran the experiment in the same way as the experimental group in all other aspects.

*No light group*: In the no-light control group, animals ran the experiment as all other groups but received no light stimulation.

**Head direction analysis**. The analysis was performed using custom python scripts. To determine whether light stimulation precisely targeted to a particular window of angles, we calculated the mean resultant vector length for the distribution of stimulated angles, which measures the concentration of angles in a distribution. Lengths vary between 0 (the underlying distribution is uniform) to 1 (all angles in the underlying distribution are exactly the same). Thus, for stimulated angles, we expect non-zero lengths close to 1. It is possible that the distribution of stimulated angles could be determined simply by a bias in the animals' behavior (i.e., the animal by chance always faces the direction we have chosen as the target window). To test against this possibility, we generated null distributions by randomly sampling angles from the full distribution of angles explored by the animal. The number of samples was set to equal the number of stimulation angles. Angles were randomly sampled in this way 1000 times, and each time a mean resultant vector length was calculated. The null distribution comprised the 1000 means (note that null distributions are centered near 0). For each session, the resultant mean vector length was well above a 99% cutoff of the null distribution, indicating that our stimulation angle precision was a result of accurate posture detection rather than a bias in animal behavior.

**Imaging of brain sections**. Brain sections were DAPI labeled (0.2 μg/ml) and overview images were acquired using a widefield microscope (Zeiss AxioScan.Z1). Based on the overall expression and fiber placement, selected sections were additionally imaged with a spinning disk microscope (VisiScope CSU-W1). Acquired z-stacks were used for quantification using FIJI[60]. Selection criteria for the quantification of Cal-Light labeling included the correct placement of the fiber ferrule above the target region as well as an injection (Supplementary Fig. 2). Mice that did not match the criteria were only included in the evaluation and quantification of DLStream performance.

**Experimental setup**. The corresponding arenas were placed in a closable compartment with isolation from external light sources. A light source was placed next to the setup so that the arena was evenly lit. The camera was placed directly above the arena. During experiments, the compartment was closed to minimize any disrupting influences from outside. All devices were triggered using NI 6341 data-acquisition board (National Instruments Germany GmbH, Munich) in combination with the Python *nidaqxm* library connected via USB 3.0 to a PC (Intel Core i7-9700K @ 3.60 GHz, 64 GB DDR4 RAM, and NVidia GeForce RTX 2080 Ti(12GB) GPU). For all experiments, we used the Intel Realsense Depth Camera D435 (Intel Corp., Santa Clara, CA, USA) at 848 × 480 and 30 Hz to enable reliable streaming at all times. Although the webcam is capable of 60 Hz and higher resolution, we found that these settings gave reliable framerate and the optional addition of depth data.

We have successfully installed and tested DLStream on Windows 10 and Ubuntu 18.04.05 OS. DLStream was developed in the open-source programming language Python. Python includes open-source libraries for most available devices or desired functions, which allows DLStream to utilize and control a wide range of devices. Virtually any webcam/camera can be used with any framerate and resolution considering hardware requirements and limitations.

**Hardware latency and detection accuracy during optogenetic stimulation**. The latency between posture detection and optogenetic stimulation was estimated by manually annotating videos of sessions from three different mice. For this, the recorded video was analyzed frame-by-frame and the frames between the event start (posture detection leading to stimulation) taken from the table-based output file and the visible onset of the laser in the video was counted. To evaluate the false-positive detection rate during experiments, we manually annotated all stimulation events during the above sessions. A detection was counted as false-positive when the annotator judged the posture of the animal (head direction) not inside the head direction window at the exact time of detection. Note that the accuracy of the pose estimation network is a major source of false detection, however, inaccurate event definitions can also lead to unintended stimulation events. Additional training of the network can increase the accuracy of the triggered stimulation.

**Reward delivery and acoustic stimulation**. The liquid reward was delivered via a custom-built reward delivery system using a peristaltic pump (Takasago Electric, Inc.). A nozzle connected to the pump was placed in the center of the northern arena wall (where the screen was located). The animal was briefly habituated to the reward during handling before continuing with habituation to the delivery system. For this, mice were first habituated to the arena and then received pretraining for reward consumption for 3 days, where they were presented with a reward at random time points. The liquid reward consisted of diluted sweetened condensed milk (1:10 with

Aqua dest.) and was delivered in a volume of ca. 4–6 µl. If not collected, the reward was withdrawn again. The aversive tone (ca. 100 dB) was delivered via a custom build piezo alarm tone generator. The device was placed above the arena.

**Pose estimation using DLC**. In all experiments, we used 3-point tracking to estimate the position, direction, and angle of the animal using the head, neck, and tail root as body parts of interest. Networks were trained using the DLC 1.11 framework. First, 300 images of relevant behavior in the corresponding arena were annotated and 95% were used for DLC network training sets. Note that for some cases, a small number of test images (5%, 15) might require further evaluation of the trained network to guarantee sufficient accuracy and generalization. Second, we used a ResNet50-based neural network[33,34] with default parameters for 500k number of training iterations and evaluated network performance. For each experiment type, a different network was trained using the same approach.

For benchmarking DLStream's upper-performance limits, we used 300 labeled images with relevant behavior (95% training set) labeling either 9 or 13 body parts. The same training set was used to train several neural networks based on different architectures or depths (ResNet50, ResNet101[33,34], MobileNetv2[35]) available through the DLC 2 framework with default parameters for 500k number of training iterations. After training, the networks were benchmarked within DLStream using a DLStream function (python deeplabstream.py --dlc-enabled --benchmark-enabled) with 3000 consecutive frames. Data were collected and average framerate, as well as standard deviation, was calculated for 4 different image resolutions (1280 × 1024, 640 × 512, 416 × 341, 320 × 256) available to the Basler acA1300-200um camera (Basler AG, Germany), which acquired frames at a rate of 172 Hz.

**Posture detection in DLStream**. We extracted the raw score maps from the deep neural network analysis and used them for posture detection. First, body part estimation, similar to the DLC approach, was conducted by local maxima detection using custom image analysis scripts. The resulting pose estimation was then transferred into postures. For this, each possible combination of body parts was investigated and filtered using a closest distance approach. DLStream detects estimated postures and compares them to relevant trigger modules for closed-loop control of experiments. To evaluate our own DLC trained networks, we measured the pose estimation error and compared it to a human-labeled data set (labeled by a single human annotator). For this, we extracted a new image set from our optogenetic experiment sessions ($n = 597$) and measured the average difference (Euclidean distance) between human annotation and pose estimation in position, as well as resulting head direction angle. Additionally, we calculated the false-positive/false-negative rate of hypothetical head direction triggers with differently sized angle windows (60, 50, 40, 30, 20, 10). To counter any non-uniform distribution of head direction angles, we averaged the rates for multiple ranges per bin (e.g., 0–60°, 60–120°, 120–180°) and calculated the standard deviation. See Supplementary Fig. 3 for details.

**DLStream output and adaptability**. DLStream is storing posture detection and information from experiments in a table-based file (see also Supplementary Data 3) that can be opened by any standard system. The file is indexed by the frame ID from the camera stream and provides information on the estimated position of all tracked body parts, the status of the experiment (whether it is active or not), and a "trial" column which is used to give event/trial-specific information during experiments (e.g., negative or positive trial during conditioning or stimulation active/not active during optogenetic experiments). The table also includes a "time" column where experimenters can see the exact inference time between each frame and the actual time that passed during the experiment.

DLStream experiments are not limited to the body parts used in our experiments and can utilize any combination of pose estimated body parts. DLStream's posture detection is stored as a "skeleton" (a set of named body parts) which is directly taken from the DLC network. Each body part or a set of body parts can be selected for the design of user-defined experiments.

DLStream users are not limited to the triggers and experiments used for the experiments in this paper but can either use provided modules or design their own modules with the help of our in-depth tutorials (https://github.com/SchwarzNeuroconLab/DeepLabStream). Currently available triggers include speed (Supplementary Movie 1), head direction, and ROI-based detection.

For a further step-by-step explanation, we included a guide on our GitHub page.

**Statistics and reproducibility**. Paired *t*-tests were used for statistical comparisons of data. All data presented in the text are shown as the mean ± standard deviation. Uncorrected alpha (desired significance level) was set to 0.05 (* <0.05, ** <0.01, *** <0.001). Sample sizes and numbers are indicated in detail in each figure caption and main text. Exclusion criteria, if applied, are specified in each corresponding method section.

**Reporting summary**. Further information on research design is available in the Nature Research Reporting Summary linked to this article.

## Data availability

The data that support the findings of this study are available from the corresponding author upon reasonable request.

## Code availability

DLStream[56] is available to the scientific community under https://github.com/SchwarzNeuroconLab/DeepLabStream. Tutorials and further information on how to use and adapt DLStream is available under the same address.

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

## Acknowledgements

We would like to thank Jonathan Ewell for language editing and feedback on the manuscript. We also want to thank Liubov Sokhranyaeva for assistance in establishing the Cal-Light system and Lina Zabawa for assistance in data visualization and performance testing. Work was supported by the DFG, SFB 1089, SPP 2041 to M.K.S., and VW Stiftung Freigeist fellowship to L.A.E.

## Author contributions

Conceptualization, J.F.S. and M.K.S.; methodology/experiments, J.F.S., M.L., L.K.; code development, J.F.S. and M.L.; experimental design, J.F.S., I.P., L.A.E., and M.K.S.; writing/reviewing/editing, J.F.S., L.A.E., and M.K.S.; supervision, M.K.S.; funding acquisition, M.K.S. and L.A.E.

## Funding

## Competing interests

The authors declare no competing interests.
