## [Peer Review File · Communications Biology]

Reviewers' comments:

Reviewer #1 (Remarks to the Author):

Summary:

The authors present a novel analysis tool, named DeepLabStream, which utilises real-time automated markerless pose estimation, based on trained DeepLabCut architectures. Instead of inferring poses offline on captured video recording, the authors aim to analyse the video stream of a webcam continuously to trigger events based on changes in posture. While only static events, such as head and body orientation are used as cues in their system, the exciting potential to also enable time-variable cues, such as movement speed or simple action recognition, are briefly discussed. As illustrative examples of the applicability of their proposed pipeline, the authors present two studies: (1) automated second order conditioning, and (2) optogenetic labelling of neurons, both tied to the head orientation of mice in a closed arena.

Given my background in machine learning and computer vision as well as their application in behavioural studies of insects, my knowledge of optogenetics is limited. Therefore, my review is focused on implementation, novelty, and versatility of the developed computational approach.

Overall Impression:

In general, the study presents a promising and attractive approach for future behavioural studies, with plenty of potential to spark further innovation in novel experimental designs for behavioural research. Overall, the manuscript is quite well written. While there is clearly novelty in the development of a "general-purpose pipeline" for the control of behavioural experiments based on real-time pose estimation, a few aspects of this study, especially regarding the developed software, are not yet ideally presented and require clarification before I can recommend publication.

A key novelty of the presented system is the possibility for to conduct dynamic feedback between postural changes and experimental conditions (in an ideal case, a true "closed-loop" experimental design). However, the latency between observed behaviour and response of the developed system is currently not reported, rendering it difficult to judge the quality of its implementation, and indeed at what time scale the system can achieve closed-loop feedback. Without further information on latency, it is impossible to judge for which types of studies the presented solution is suitable. I appreciate the dependence on used hardware, but even more so, a recommendation should be made for specific systems capable of fully utilising the developed system.

There are currently two major shortcomings in the description and justification of the described approach: (1) Throughout the manuscript the authors repeatedly mention the potential for the extraction of time dependent triggers (e.g. movement speed, action recognition), yet do not discuss how those would be implemented, and indeed they only used static cues in their proof-of-concept studies. If the authors claim applicability of their tool for non-static cases as well, this should be shown experimentally. Otherwise, while presenting a solution for posture-based triggers, the claim of a truly closed-loop system is not sufficiently strong, as the possibilities for dynamic interactions with the studied animal(s) are limited.

(2) The performance of the developed approach needs to be reported more thoroughly. Using only three landmarks during poster estimation reduces inference times, but as many experiments likely require a larger number of markers or even multiple animals, it would be important to report on the influence on processed frames per second, especially, when cues become computationally expensive. Furthermore, it should be clearly stated which architectures exactly were used, as the DeepLabCut framework natively supports several feature-extractor networks of different depths.

How did choice of architecture influence performance of DeepLabStream?

While some aspects of the computational side need further clarification, the overall presented approach appears novel and promising. To fully evaluate the developed approach and judge its versatility I would like to have access to the written code in its current stage. Structurally, a few sections show redundancies and should be shortened.

Specific comments:

- Abstract:

1. The general notion of using these tools in real-time to influence and study behaviour through triggered interactions is a great concept but potentially not ideally communicated in these first lines. It might help the reader understand what you have achieved specifically within the scope of the study if "what can be done" and "what has been done" are separated more clearly. I would advise to first elaborate on the system's potential use and then on your concrete application and results, rather than alternating back and forth.

2. The claim that real-time processing of posture data to correlate with neuronal activity is necessary for the study of behaviour, is not adequately justified here. For example, both case studies presented by the authors could have been executed just as well by an observer monitoring a video stream. As there is no information provided on the latency between a captured frame and a processed response, it is impossible to tell whether the system could provide a better response than a human experimenter.

- Introduction:

3. When introducing robust pose-estimation for behavioural studies, Graving et al, DeepPoseKit should also be cited. (Done in the discussion but not initially in the introduction)

4. (very minor correction) Conventionally, in deep learning the term "deep neural network" is preferred over "deep neuronal network"

5. I recommend restructuring the first paragraph. As the motivation of the presented research is problem-driven, the introduction would benefit from first outlining key questions / limitations of the field, before mentioning recent advancements and state-of-the-art methods.

- Results:

6. It would be advantageous to report the latency and achievable framerate during monitoring as opposed to later offline analysis when first introduced in the results section. This would provide a frame of reference what low vs. high framerates indicate in that context

7. Fig. 2) While the module-based system should be intuitive, the schematic representation is a little difficult to follow. While the function of each module is outlined briefly, the addition of conditionals is not entirely clear: Are conditionals descriptions of how modules function in combination, or are they additional elements that need to be defined by the experimenter?

8. Fig. 2) In example (e), the timer module is tied to the removal of the presented reward if the required action has not been carried out by the animal, whereas in (a) and (b), it is only used as a delay. For both points 7. and 8., additional clarifications should be provided in the figure captions.

9. Fig. 3) The "yellow triangle" depicting the viewing angle is never used. I understand that the grey and green shaded areas indicate the viewing angle, depending on whether a stimulus is presented, but it is somewhat confusing to have the legend reference something not found in the

figure itself

10. Fig. 3) There also does not appear to be a “red square” used anywhere in the figure to indicate the previous reward location. For both points 9. and 10., either alter the figure or provide additional details within the figure caption.

11. Fig. 3) I would recommend including short titles for a), b), c) within the figure itself, so the reader can immediately tell, what stage of the conditioning task is being shown.

12. Fig. 3) (d) mentions “pretraining” but there is no explanation of what this phase refers to, or how it is different from habituation.

13. I appreciate that the actual delay observed between camera capture and output will vary depending on the used setup, thus, a multitude of components, but I believe that it is crucial nevertheless to report this number, as an example. While the reported times for the processing of video frames appears promising, yet not dissimilar to the reported times by a standard implementation of DLC, this provides no insight into how much time passes between the acquisition of an image and the response of the proposed pipeline. The overhead associated with processing a continuous video stream, as opposed to a contained video clip or array of images, may additionally influence the time required to analyse each frame.

- Discussion:

14. The initial statement regarding the general design of pose-estimators being created for offline applications is not entirely accurate. Especially in human pose-estimation, there are plenty of frameworks designed for online and real-time applications. See for example:

a. XNect: Real-time Multi-Person 3D Motion Capture with a Single RGB Camera Dushyant Mehta et al. 2019

b. Human Pose Simulation and Detection in Real Time Using Video Streaming Data, Prasanth Kumar Ponnarassery et al. 2020

c. Real-Time Skeleton-Tracking-Based Human Action Recognition Using Kinect Data, Georgios Th. Papadopoulos et al. 2014

15. “Speed” is mentioned as an example trigger, yet, the system as it is presented, only supports single-frame postural information as triggers. In general, it should be elaborated how sequences of actions / postures are to be treated and processed, as well as accompanying implications for changes in processing times, when the application of filters become important.

16. The distinction between “task” and “activity” is somewhat vague. Clear definitions for both need to be provided.

17. As actions can seldomly be encapsulated by a single frame, action recognition should be coupled with the extracted posture. The authors need to elaborate on how action recognition would be implemented in their approach and where potential difficulties lie.

18. Supplement fig 3 does not seem to be available! (referring to bottlenecks in processing speeds)

19. Without knowing more about the implementation of DLStream, the comparison to similar work regarding framerate but not response time without hardware specifications gives little insight into the comparability between the two systems (Forys et al. 2020). As mentioned in (13.), these details are crucial.

20. Reference to additional works regarding multiple animal tracking requires citation.

- Code availability:

21. At the moment, the link is not accessible, therefore it is not possible to judge the claims made about the versatility or limitations of the created software. Please provide your written code for the continuation of the revision process.

- Methods:

22. (Conditioning task) Please provide a number for the number of animals that correctly learned the association in the negative trial.

23. Head direction-dependent optogenetic simulation and Action dependent light simulation share a lot of information. Consider combining the two sections

24. (Experimental Setup) If the depth data was not used, why were higher framerates not considered? With the argument that pushing the framerate further to test the upper limits of the online DLC network, this would have made a better comparison to Forys, et al. possible

25. Has there been any testing regarding fewer or more tracked body parts? An increase in network complexity for a larger number of score maps might influence inference time significantly.

26. It is unclear, whether modules of "possible body parts" need to be custom built for comparison of current states of the animal. It is important to elaborate on how a user of the developed system would introduce new cues / modules.

- Minor language corrections:

27. P11, "The screen was the same size as the arena wall it was placed at."

28. P11, "Directly after the surgery animals were administered 1 ml 5% Glucosteril solution"

Reviewer #2 (Remarks to the Author):

The manuscript under review by Schweihoff et al titled "DeepLabStream: Closing the loop using deep learning-based markerless, real-time posture detection" provides an open-source platform for autonomous real-time pose-estimation (via DeepLabCut) and close-loop behavioral experimentation. Specifically, the authors develop and provide a Python-based pipeline that is validated using two behavioral procedures, in (1) a second-order olfactory conditioning task and (2) an opto-tagging head direction task.

Comments:

1) DeepLabCut is a very popular, and highly used, pose-estimation platform. However, there are numerous others (and surely more to be developed in the near future), and it would strengthen the use and general applicability of this pipeline if it were validated with other pose-estimation architectures like DeepPoseKit, LEAP (or SLEAP) and also the more recent maDLC release. At the least, a discussion on how DeepLabStream (abbreviated DLS from here) can, or cannot, be used with these other architectures should be included. If there are technical reasons DLS will only work with DLC, this should be explicitly explained.

2) Is DLS only compatible with single-animal experiments? Can multiple (dissimilar) looking animals be pose-estimated concurrently, or similar looking animals with maDLC? I imagine the addition of more pose-estimation features may lead to latency. How many features can be pose-

estimated on a single, or multiple, animals before latency becomes too challenging to work with?

3) More information on the table-based file (mentioned on page 2) would be useful, as understanding the dataframe can often be the crux of adopting and adapting an open-source package. This deserves at least some mention in the Methods.

4) 30FPS is adequate for many, but not all, behaviors as noted in the discussion section. The bottleneck is likely determined by the resolution of the captured video. A parametric dataset using resolution downsampled comparisons to the original 848x480 video would be very helpful in understanding how DLS could be adapted to behaviors that require higher sampling. Notably, a lot of behaviors are recorded at lower resolutions, and DLC can pose-estimate lower resolution images just fine. Would be useful information for users!

5) Minor comment, but the github link provided in Code Availability on page 7 leads to a 404 due to the hyphen.

Overall, I think this protocol will be useful to the field and is developed to a point that is suitable for publication.

Reviewer #3 (Remarks to the Author):

Schweihoff et al present DeepLabStream, a system for real-time tracking and stimulus delivery using DeepLabCut, and demonstrate its performance in two example experiments. Such a tool has many potential applications in systems neuroscience, and the authors' efforts to create and share such a tool with the community is laudable. However, the impact of the paper is somewhat reduced by the two example experiments provided, where issues of experimental design make it hard to know how well DLStream is actually working. A simpler example experiment (eg classical conditioning or conditioned place aversion) would have made a much more convincing case that the system works as advertised. In addition, a more rigorous evaluation of DLStream's performance is needed. More specifically, I have the following comments:

The evaluation of DLStream's performance is the most important part of the paper, but it is currently a bit limited. A few points on this topic:

1) The authors characterize the accuracy of DLStream by computing the average Euclidean distance between nose, neck, or tail points on consecutive frames, as well as the average head direction variance between frames. As the authors themselves note, these values depend on two terms-- the actual movement of the animal during the experiment, and the performance errors of the pose estimate. Thus there is no way for the reader to determine from the reported Euclidean distances/head directions how accurate the system's pose estimations really are. Because of this, I believe that it is necessary for the authors to collect a novel, manually annotated "test set" of nose/head/tail point locations, and to evaluate DLStream accuracy with respect to this test set.

2) While the *average* error of pose estimation may be low, what is more important is how often DLStream produces false positive/false negative event detections. This could be computed by, for example, having a human who is blind to DLStream output manually annotate when the mouse is performing the target behavior (eg looking at the screen or smelling a petri dish in the conditioning experiment, or facing the target direction in the Cal-Light experiment), and comparing the accuracy of DLStream to these manual annotations. While time consuming to collect, it is critical that these performance measures are established, or the reader will be left with no way to evaluate how usable DLStream might be for their own work.

3) It would also be helpful to see how the above performance depends on the trigger used: for example, in the Cal-Light task, what is the false positive/negative rate as a function of the stimulation angle around the target direction? (eg at 60 degrees, 50, 40, 30, 20, 10, 5...)

4) The paper does not explain how the end user constructs triggers- is there an API or user interface? The paper needs a section to explain, step-by-step, how triggers are actually created by an end user of the system, and what conditions can be used to create a trigger. This is ostensibly done in the Discussion, however only in broad terms, with no references to how these actions are supported by the DLStream software/interface.

5) The Discussion also makes contradictory statements about what kinds of triggers an end user can build. One sentence states "Out of the box, DLStream supports triggers based on single-frame postural information, although posture sequences or complex behavior modules are also possible once behavior based on collected posture data has been classified, modeled, and integrated as trigger modules into DLStream." This makes it sound like a potential end-user of DLStream would only have access to single-frame posture triggers, however earlier in the paragraph the authors state "With DLStream, a combination of triggers based on the animal's posture or posture sequence is now possible." In general, the authors should tone down the "sales pitch" style language in their introduction and discussion, and state the functionality and limitations of their system directly.

6) I am surprised that only 300 frames were used to train DLC: while DLC advertises that it can be trained using a small number of examples, my experience in talking to labs that use it is that in practice many more (a few thousand) frames are needed to achieve satisfactory performance. Have the authors confirmed that DLC performance has plateaued with only 300 training examples? Might more training data further improve performance of the system?

7) It is unclear from the text whether DLStream is compatible with other choices of tracked body parts (instead of the nose-neck-tail points tracked here.) Is this the case? If so, how is the response time of DLStream affected by the number of body parts tracked?

I also had the following comments on the two experimental use-cases presented:

8) The results of the second-order conditioning task are a little unusual. As noted in the text, the strongest effect present in Fig 3E seems to be that the mice are spending more time in the quarter of the arena where they previously received a reward. However, I don't see evidence for a "clear odor preference" as claimed in the discussion: none of the plots show that the mice investigate the Vanillin more than either of the two novel odors, nor do the mice spend less time investigating the Rose dish than the novel odors. The odor location plots do show a statistically significant difference between Vanillin and Rose investigation times, but the confound of the reward port location makes the plots hard to digest. The fact that the two groups show very different investigation times towards Acetophenone also makes me distrust the odor preference results.

9) While there's a lot going on in this experiment (re odor location/reward location interactions), there's little in the text that attempts to unpack these complexities of the data and justify the claim of a learned odor preference-- the authors instead focus on their finding of a preference for the reward corner. While this is a valid finding, it is problematic for this paper because preference for the reward corner is a behavior that would emerge whether or not the DLStream system was actually working. Would it be possible to repeat this experiment with the reward port in a location equidistant to the two odor sources (eg at the center back of the arena) so that the performance of DLStream could be evaluated without having to worry about this added confound of reward location?

10) There are several design decisions in the Cal-Label experiment that seem strange. In Figure 4c, the red vs gray portions of the right panel seem to suggest that light stimulation was only

actually delivered for a tiny fraction of the time that the tracked mouse's head was pointed in the target direction (blue arc). Is this all because of the 15-second inter-stimulus time? This extreme delay is not commented on in the main text, however the methods cite that this is "to avoid overheating of brain tissue". This seems strange- other groups have delivered light stimulation for much more than 5 seconds without issue. If overheating is truly a concern, why not reduce the power, stimulation frequency, or duty cycle of the laser? Or if all this is not possible- why not narrow the target direction to less than 60 degrees?

11) It also seems like light stimulation is preferentially delivered when the animal's head is between \sim -30 (ie 330) and 0 degrees, with much less stimulation between 0 and 30 degrees. Why this difference?

12) In figure 4e, the legend claims this is a heatmap of "the relative occupancy of the mouse within the arena during the session", however text claims 4e shows occupancy only during times when stimulation was delivered. Which is it? Ideally you should show both occupancy during the session and occupancy during stimulation, as well as the ratio of the two (ie the conditional probability of stimulation given occupancy). Also, the text states "light stimulation was independent of the animal's position in the arena"—what is the basis of this claim? 4e shows a pretty dense occupancy peak at the arena wall, just to the left of the stimulation zone (dark red), suggesting that the animal is in fact preferentially stimulated when it is in this location. This could occur if the animal spends most of the time circling the edges of the arena in a fixed direction, rather than exploring randomly. Perhaps placing a few small food rewards throughout the arena could encourage more varied behavior.

13) I do not see any data that validates that the cells labeled by Cal-Light are actually tuned for the target head direction- all that's shown in Figure 4 is evidence that some cells were labeled. This is not acceptable; the reader has no way of knowing whether the experiment worked at all. The text claims that the Yoked control mice do not have labeled cells, but no data is shown to support this claim, and this is anyway not significant evidence that head direction cells were labeled in the Experimental group.

Other minor points

1) The legend for 3C mentions a "red square" but I do not see a red square in the figure. (Also the "yellow triangle" mentioned in the legend for 3A is not shown in the figure, but I am guessing that the gray triangles are what is meant here.)

2) Check Figure 4 legend- some panel lettering is incorrect, and panels h/i legends are possibly incorrect/missing.

Response to the referees:

We thank the three critical reviewers for the time they took to thoroughly assess our manuscript. We also thank them for their constructive comments on how to further improve our manuscript, which we addressed in the following point-by-point response (in red). For a general overview of collective changes made in the revised version of the manuscript we provide a summary table (please see below).

Table 1: Short summary of changes in the revised manuscript

Figures & Tables	Figure 1	Minor correction in the figure description
	Figure 2	Minor changes to panel a+b and additional information in figure description
	Figure 3	Minor Changes to panels a-c (a-c : additional titles; a : position of reward port adjusted (blue drop); b : figure legend (viewing angle/triangle) color adjusted); Major changes to panels c+e (Experiment was repeated, c : arena outline adjusted to rectangle and odor position updated, d : removed old panel and added new bar graph with individual data points, e : removed panel completely to reduce complexity of figure); Changes to figure description (c-e)
	Figure 4	Panel e is now e+f, which results in panels f-l now being panels g-j in the revised figure. Additional plot/heatmap in revised panel f; Minor changes to the figure description (updated lettering and split section e to e+f with updated information on the new plot)
	Suppl. Figure 1	Additional panel c now includes results of counting yoked group, updated panel b with a more representative image set; Additional information in the figure description for the new panel c
	Suppl. Figure 3	New suppl. Figure showing evaluation of false positive/negative rate of event detection in a human labelled dataset
	Suppl. Figure 4	New suppl. Figure showing additional examples of head direction angles during optogenetic light stimulation (like Fig. 4 c)

	Suppl. Table 1	New suppl. Table showing additional performance tests and benchmarks of DLStream
Text	Abstract	Restructured abstract based on comments from Reviewer #1 and #3
	Introduction	Restructured introduction based on comments from Reviewer #1
	Results	Includes several changes based on requests from the reviewers. Additional sections reporting the performance in greater detail (also see new suppl. figures/tables)
	Discussion	Includes several changes and additions based on requests from the reviewers. Updated discussion to fit to new results from the repeated experiment.
	Methods	Minor changes based on reviewer requests. Additional information and sections based on reviewer requests. Updated sections to fit repeated experiment

Reviewers' comments:

Reviewer #1 (Remarks to the Author):

Summary:

The authors present a novel analysis tool, named DeepLabStream, which utilises real-time automated markerless pose estimation, based on trained DeepLabCut architectures. Instead of inferring poses offline on captured video recording, the authors aim to analyse the video stream of a webcam continuously to trigger events based on changes in posture. While only static events, such as head and body orientation are used as cues in their system, the exciting potential to also enable time-variable cues, such as movement speed or simple action recognition, are briefly discussed. As illustrative examples of the applicability of their proposed pipeline, the authors present two studies: (1) automated second order conditioning, and (2) optogenetic labelling of neurons, both tied to the head orientation of mice in a closed arena.

Given my background in machine learning and computer vision as well as their application in behavioural studies of insects, my knowledge of optogenetics is limited. Therefore, my review is focused on implementation, novelty, and versatility of the developed computational approach.

Overall Impression:

In general, the study presents a promising and attractive approach for future behavioural studies, with plenty of potential to spark further innovation in novel experimental designs for behavioural research. Overall, the manuscript is quite well written. While there is clearly novelty in the development of a “general-purpose pipeline” for the control of behavioural experiments based on real-time pose estimation, a few aspects of this study, especially regarding the developed software, are not yet ideally presented and require clarification before I can recommend publication.

A key novelty of the presented system is the possibility to conduct dynamic feedback between postural

changes and experimental conditions (in an ideal case, a true “closed-loop” experimental design). However, the latency between observed behaviour and response of the developed system is currently not reported, rendering it difficult to judge the quality of its implementation, and indeed at what time scale the system can achieve closed-loop feedback. Without further information on latency, it is impossible to judge for which types of studies the presented solution is suitable. I appreciate the dependence on used hardware, but even more so, a recommendation should be made for specific systems capable of fully utilising the developed system.

Many thanks for this positive and constructive evaluation. According to the reviewer’s suggestions, we have thoroughly addressed and clarified raised concerns regarding DLStream software solution, response latency of the system and have recommended additional suitable systems that can be utilized by our system (for detail please see below).

There are currently two major shortcomings in the description and justification of the described approach:

Throughout the manuscript the authors repeatedly mention the potential for the extraction of time dependent triggers (e.g. movement speed, action recognition), yet do not discuss how those would be implemented, and indeed they only used static cues in their proof-of-concept studies. If the authors claim applicability of their tool for non-static cases as well, this should be shown experimentally. Otherwise, while presenting a solution for posture-based triggers, the claim of a truly closed-loop system is not sufficiently strong, as the possibilities for dynamic interactions with the studied animal(s) are limited.

We thank the reviewer for this constructive criticism. Indeed, the current version of the manuscript uses only static cues and does not discuss the extraction and implementation of time dependent triggers.

In general, DLStream is running without batch processing multiple images, but with frame-by-frame analysis via DeepLabCut (DLC) trained networks. However, any number of extracted pose estimations can be stored within DLStream (e.g. in the experiment or trigger module) and be used for posture sequence-based triggers. As correctly pointed out by the reviewer, we did not actively show an experiment with such a trigger as both use cases can be sufficiently solved by using single-posture triggers.

To now show the potential of DLStream to utilize sequence (time)-dependent triggers we additionally released a “speed-dependent trigger” and released extensive tutorials/guides on the GitHub page to help potential users designing their own triggers or adapt existing ones to their needs (<https://github.com/SchwarzNeuroconLab/DeepLabStream/wiki/Introduction>).

As a proof of concept, and to truly claim closed-loop applications we now added a Supplementary Video showing a posture-sequence dependent trigger in the revised version of the manuscript (Suppl. video 1).

Generally, as to our knowledge for action recognition and dynamic interactions, as raised by the reviewer, classification-based triggers are the best solution. In the current version of the manuscript we also discuss this challenge (Discussion Page 6, lines 3-17).

(2) The performance of the developed approach needs to be reported more thoroughly. Using only three landmarks during pose estimation reduces inference times, but as many experiments likely require a larger number of markers or even multiple animals, it would be important to report on the influence on processed frames per second, especially, when cues become computationally expensive. Furthermore, it should be clearly stated which architectures exactly were used, as the DeepLabCut framework natively supports several feature-extractor networks of different depths. How did choice of architecture influence performance of DeepLabStream?

While some aspects of the computational side need further clarification, the overall presented approach appears novel and promising. To fully evaluate the developed approach and judge its versatility I would like to have access to the written code in its current stage. Structurally, a few sections show redundancies and should be shortened.

We agree with the referee that the manuscript would greatly benefit from a broader performance report and now included additional data on a variety of frame-resolutions (320x256; 416x341; 640x512; 1280x1024), several different network architectures (ResNet50, ResNet101, MobileNetv2) as well as body part numbers (3, 9 and 13) in the revised version of the manuscript (new supplementary table 1, page 8; see also below).

In addition, we added the following paragraph to the results in the revised manuscript (Page 5, Line 26-34):

“Second, we tested a different camera (Basler acA1300 – 200 um), which lacks the depth capabilities of the Intel RealSense camera but comes with increased framerate, on the advanced configuration with different image resolutions (ranging from 1280x1014 to 320x256 px) to benchmark DLStream’s upper performance limits with more standardized cameras and resolutions. While we initially used DLC trained ResNet50^{33,34} networks during experiments, we additionally evaluated the capabilities of the other available models (ResNet101^{33,34}, MobileNetv2³⁵) and also a higher number of body parts (3, 9 and 13 body parts). In our hands, DLStream’s latency reached a maximum of 130 ± 6 Hz (ca. 8 ms) with the MobileNetv2 architecture at 320x256 px resolution, while the ResNet50 network reached its upper limit at 94 ± 6 fps (ca. 10 ms) at the same resolution (see Supplement Table 1 for more details).”

We also cited the recent work from the original DLC authors that test the inference speed of available networks (Mathis et al. 2018, Kane et al. 2020).

As suggested by the reviewer, we now included a more in-depth description of our network choice in the method section (Page 15, Line 39-51). We hope that our modifications now allow potential users to evaluate their configuration choice when implementing DLStream into their experiments.

As indicated by the reviewer, the choice of architecture in combination with the number of tracked body parts and the input frame resolution greatly influences the available inference speed of DLStream. Currently our fastest tested solution is the MobilNetV2 trained for three body parts with an input resolution of 320 x 256 px. (Please also see the added section in the revised manuscript (Page 5, Line 26-34).

We regret that the reviewer was not able to access the code due to an inaccuracy on our side during submission of the manuscript. The formatting added a line-break (with a hyphen) in the middle of the link, which we did not notice. We apologize again, fixed the hyperlink (<https://github.com/SchwarzNeuroconLab/DeepLabStream>) in order to allow fast access to the current version of the code.

For the reviewer’s consideration, we included the new table underneath:

New Supplement Table 1 | Performance of different network architectures in DLStream in relation to number of estimated body parts and image resolution.

Network	Resolution	3 Body parts	9 Body parts	13 Body parts
		Average FPS	Average FPS	Average FPS
MobileNetv2	320x256	164.04 +/- 7.28	130.55 +/- 6.51	79.29 +/- 19.18
	416x341	119.73 +/- 8.42	86.64 +/- 3.43	67.50 +/- 10.95
	640x512	60.51 +/- 2.01	54.24 +/- 0.94	46.76 +/- 2.91
	1280x1024	16.61 +/- 0.26	16.19 +/- 0.20	14.99 +/- 1.14
ResNet50	320x256	107.58 +/- 8.68	94.30 +/- 6.21	67.01 +/- 10.36
	416x341	79.52 +/- 3.00	66.70 +/- 1.83	55.49 +/- 4.99
	640x512	44.92 +/- 1.44	41.03 +/- 0.52	36.50 +/- 1.88
	1280x1024	13.61 +/- 0.35	13.25 +/- 0.12	12.32 +/- 0.86
ResNet101	320x256	68.29 +/- 2.23	64.72 +/- 1.86	60.35 +/- 6.13
	416x341	54.85 +/- 1.51	49.99 +/- 0.94	48.34 +/- 3.01
	640x512	32.05 +/- 0.51	30.47 +/- 0.33	30.18 +/- 1.88
	1280x1024	9.80 +/- 0.28	10.33 +/- 0.34	9.92 +/- 0.76

Specific comments:

- Abstract:

1. The general notion of using these tools in real-time to influence and study behaviour through triggered interactions is a great concept but potentially not ideally communicated in these first lines. It might help the reader understand what you have achieved specifically within the scope of the study if “what can be done” and “what has been done” are separated more clearly. I would advise to first elaborate on the system’s potential use and then on your concrete application and results, rather than alternating back and forth.

We have carefully reviewed the abstract and tried our best to modify the revised abstract according to the reviewer’s suggestions. Additionally, and in line with comments from reviewer #3 we toned down our “sales pitch” language and refrained from claims not supported by experiments.

Revised abstract:

“In general, animal behavior can be described as the neuronal-driven sequence of reoccurring postures through time. Most of the available current technologies enable offline pose estimation with high spatio-temporal resolution. To correlate behavior with neuronal activity it is often necessary to detect and react to behavioral expressions in real-time. Here we present DeepLabStream, a versatile closed-loop tool providing real-time pose estimation to deliver posture dependent stimulations. DeepLabStream has a temporal resolution in the millisecond range, can utilize different input, as well as output devices and can be tailored to multiple experimental designs. We employ DeepLabStream to semi-autonomously run a second-order olfactory conditioning task with freely moving mice and optogenetically label neuronal ensembles active during specific head directions.”

2. The claim that real-time processing of posture data to correlate with neuronal activity is necessary for the study of behaviour, is not adequately justified here. For example, both case studies presented by the authors could have been executed just as well by an observer monitoring a video stream. As there is no information provided on the latency between a captured frame and a processed response, it is impossible to tell whether the system could provide a better response than a human experimenter.

We agree with the reviewer and changed the abstract accordingly (please see above). In addition, in the results part of the revised version of the manuscript we now provide latency measures between a captured frame and a processed response (Page 5, Line 12-16).

“We also measured the hardware latency to estimate the time between posture detection and triggered stimulation during optogenetic sessions from three different mice ($n = 164$). Here, the resulting light stimulation occurred within 5 frames (4.8 ± 1.1 frames at 30 fps; ≈ 150 ms). It is important to consider here that the total latency critically depends on the individual setups and the intrinsic parameters of connected components.”

• Introduction:

3. When introducing robust pose-estimation for behavioural studies, Graving et al, DeepPoseKit should also be cited. (Done in the discussion but not initially in the introduction)

Included as requested (Page 1 line 17).

4. (very minor correction) Conventionally, in deep learning the term “deep neural network” is preferred over “deep neuronal network”

We changed the text passages to fit the convention (Page 1 line 18, page 2 line 46, page 5 line 38+40, page 7 line 36, page 15 line 40+45+52)

5. I recommend restructuring the first paragraph. As the motivation of the presented research is problem-driven, the introduction would benefit from first outlining key questions / limitations of the field, before mentioning recent advancements and state-of-the-art methods.

We agree with the reviewer and restructured the introduction accordingly.

“A major goal in behavioral neuroscience is the correlation of behavioral expressions with neuronal activity. For best effectiveness, however, behavior should be identified in real-time, allowing for instantaneous feedback, i.e. “closed-loop” manipulation based on the current behavioral expression^{1,2}. Currently, such experimental systems often rely on specialized, on-purpose setups, including intricate beam brake designs, treadmills, and virtual reality setups to approximate the movement of the investigated animal in a given environment and then react accordingly^{3–11}.

Classic manipulations of neuronal activity such as lesions, transgenic alterations, and pharmacological injections result in long-lasting, and sometimes chronic changes in the tested animals, which can make it difficult to interpret behavioral effects. In recent years, there has been a shift towards techniques that allow for fast, short-lived manipulation of neuronal activity. Optogenetic manipulation, for example, offers high temporal precision, enabling the manipulation of experience during experimental tasks that test mechanisms of learning and memory^{12–14}, perception^{15,16} and motor control^{17,18}. Such techniques offer a temporal resolution precise enough that the neuronal manipulation can match the timescale of either behavioral expression or neuronal computation.

Recent developments in the field of behavioral research have made offline pose estimation of several species possible using robust deep-learning-based markerless tracking^{19–21}. DeepLabCut (DLC)²⁰, for example, uses trained deep neural networks to track the position of user defined body parts and provides offline motion tracking of freely moving animals. Additionally, sophisticated computational approaches have allowed for disentangling the complex behavioral expressions of animals into patterns of reoccurring modules^{22–26}. In vivo single unit recording²⁷, along with recent advances in in vivo voltage imaging²⁸ and miniaturized calcium imaging techniques^{29–31}, facilitate real-time measurements of neuronal activity in freely moving mice. Together, these techniques provide a platform for correlating recorded neuronal activity and complex behavior.

We here introduce DeepLabStream (DLStream), a multi-purpose software solution that enables markerless, real-time tracking and neuronal manipulation of freely moving animals during ongoing experiments. Its core capability is the orchestration of closed-loop experimental protocols that streamline posture-dependent feedback to several input-, as well as output-devices. We modified state-of-the-art pose estimation based on DLC²⁰ to be able to track the postures of mice with body part precision in real-time. To demonstrate the software's capabilities, we conducted a classic, multilayered, freely moving conditioning task, as well as a head direction-dependent optogenetic stimulation experiment using a neuronal activity-dependent, light-induced labeling system (Cal-Light)¹. Finally, we discuss the versatility of DLStream to adapt to different experimental conditions and hardware configurations.”

• Results:

6. It would be advantageous to report the latency and achievable framerate during monitoring as opposed to later offline analysis when first introduced in the results section. This would provide a frame of reference what low vs. high framerates indicate in that context

We changed the structure of that section according to the reviewer's suggestion and now added additional information on achievable framerates in the revised version of the manuscript (Page 5 line 9-34).

7. Fig. 2) While the module-based system should be intuitive, the schematic representation is a little difficult to follow. While the function of each module is outlined briefly, the addition of conditionals is not entirely clear: Are conditionals descriptions of how modules function in combination, or are they additional elements that need to be defined by the experimenter?

To clarify, intentionally the combination of posture-based triggers and stimulation puzzle pieces should have been further defined by the boxes “Conditional ON/OFF” and do not hint at an additional computational element that needs to be defined by the experimenter. To facilitate an intuitive understanding of the schematic representation we removed outlines and boxes in Fig. 2 a, b and specified this in the figure legend. (revised Fig. 2 a, b, Page 2; see query # 8).

8. Fig. 2) In example (e), the timer module is tied to the removal of the presented reward if the required action has not been carried out by the animal, whereas in (a) and (b), it is only used as a delay. For both points 7. and 8., additional clarifications should be provided in the figure captions.

As suggested by the reviewer we included an adjusted description of the use of timers in the revised figure 2 (revised Fig. 2, Page 2). We added the following clarifying sentence: “For example, the timer module can be utilized to design inter-trial and -stimulus timers (b), minimum stimulation (b) or delayed triggers (e).”

For the reviewer's consideration, we included the revised figure below:

“Revised Fig. 2 | Experimental design using DLStream. a, b, Schematic design of an experimental protocol with a posture-based trigger. Manipulation can be turned “Conditional OFF” (a) and “Conditional ON” (b) based on the mouse’s behavior. The combination of several modules allows building a sophisticated experimental protocol. For example, the timer module can be utilized to design inter-trial and -stimulus timers (b), minimum stimulation (b) or delayed triggers (e). c, Description of available modules in a and b. d, Application of the above-described design in an optogenetic experiment. The stimulation is triggered dependent on head direction angle (orange arrow, α) to a reference point (red line) within target window (blue arc). e, Application of the above-described design in a classical conditioning task. The mouse is shown an image when looking at the screen (left) and the reward is removed if it does not move into the reward location within a predefined timeframe (right, green zone). The mouse’s posture is shown with orange dots.”

9. Fig. 3) The “yellow triangle” depicting the viewing angle is never used. I understand that the grey and green shaded areas indicate the viewing angle, depending on whether a stimulus is presented, but it is somewhat confusing to have the legend reference something not found in the figure itself

We thank the reviewer for noticing this graphical detail. In the revised version of the manuscript we changed the figure to only show the depicted green triangle and changed the figure text accordingly (revised Fig. 3 a-b, Page 3).

Please also note that due to a major request of reviewer #3 we repeated the entire experiment and thus adjusted significant parts of figure 3 a, c, d. Please note that the results of the modified experiment did not change the message of the figure (please also see answer to reviewer #3, query 8 and 9).

10. Fig. 3) There also does not appear to be a “red square” used anywhere in the figure to indicate the previous reward location. For both points 9. and 10., either alter the figure or provide additional details within the figure caption.

We removed the sentence mentioning a “red square” in the revised legend of figure 3.

Again, please note that due to a major request of reviewer #3 we repeated the entire experiment and thus adjusted significant parts of figure 3 a, c, d. Please note that the results of the modified experiment did not change the message of the figure (please also see answer to reviewer #3, query 8 and 9).

11. Fig. 3) I would recommend including short titles for a), b), c) within the figure itself, so the reader can immediately tell, what stage of the conditioning task is being shown.

In the revised version of the new figure 3 we included short titles (new Fig. 3 a- c, Page 3) as suggested by the referee.

12. Fig. 3) (d) mentions “pretraining” but there is no explanation of what this phase refers to, or how it is different from habituation.

To clarify this issue and to also meet the requests of reviewer #3 we removed panel d of Fig. 3 entirely, but included relevant information regarding the experimental timeline (as requested information on the pretraining phase) into the “Methods” section of the revised manuscript (Page 13-14, Line 38-10. Page 15 Line 27-35).

For the reviewer’s consideration, we included the revised figure below:

“Revised Fig. 3 | Closed-loop conditioning task. a, Conditioning. When a trial is triggered by the mouse facing the screen (left, green triangle and ring), the mouse is shown a visual stimulus (yellow lightning bolt). Mice not facing the screen do not receive the stimulus (red x). In the positive trial (middle; green lightning bolt, green line), a reward is delivered (blue drop, arrow down) and withdrawn (blue drop, arrow up) if not collected within a preset time period. In the negative trial (right; blue lightning bolt, blue line) only a loud tone (red polygon) is delivered. **b, 2nd Order Conditioning.** Upon exploration of either odor location (colored black circle) the mouse is shown one of the previously conditioned visual stimuli on the screen (left, yellow lightning bolt). Conditioning was conducted in two stages (top middle and top right). The first stage (top middle) consisted of direct contact with the odor location, while the second (top right) was dependent on the proximity of the mouse to one of the locations (black arrow) and the mouse facing towards it. **c, Odor Preference Task.** The mouse was set in an open field arena with one odor in each of the quarters (colored circles). The total investigation time of each odor source was measured. **d, Investigation time during odor preference task in odor location: ROIs encircling the odor location.** Bar graph shows STD and individual data points. $p < 0.05$ (*) paired t-test; $n = 6$ (2 trials per mouse, 1 trial excluded). V = Vanillin (S+), R = Rose (S-), VA = Valeric acid, A = Acetophenone”

13. I appreciate that the actual delay observed between camera capture and output will vary depending on the used setup, thus, a multitude of components, but I believe that it is crucial nevertheless to report this number, as an example. While the reported times for the processing of video frames appears promising, yet not dissimilar to the reported times by a standard implementation of DLC, this provides no insight into how much time passes between the acquisition of an image and the response of the proposed pipeline. The overhead associated with processing a continuous video stream, as opposed to

a contained video clip or array of images, may additionally influence the time required to analyse each frame.

We fully agree with the reviewer and added the suggested information in the revised version of the manuscript. Specifically, we estimated the latency between successful posture detection and stimulation onset for our time-sensitive optogenetic task by manually annotating three randomly selected sessions from three different mice. We counted the number of frames between the posture detection (specified in the table based output file) and the visible onset of the laser (bright blue light, visible in the video output). We included these results in the “Results” and “Methods” section of the revised manuscript (Page 5, Line 12-16, Page 15 line 16-26). In addition, we also measured the performance of DLStream with different pose estimation architectures (from DLC) and different resolutions, which showed similar results to the performance levels reported by others using real-time applications of DLC (Kane et al. 2020, Forsys et al. 2020) (new supplementary table 1, also below query #2).

• Discussion:

14. The initial statement regarding the general design of pose-estimators being created for offline applications is not entirely accurate. Especially in human pose-estimation, there are plenty of frameworks designed for online and real-time applications. See for example:

- XNect: Real-time Multi-Person 3D Motion Capture with a Single RGB Camera Dushyant Mehta et al. 2019
- Human Pose Simulation and Detection in Real Time Using Video Streaming Data, Prasanth Kumar Ponnarassery et al. 2020
- Real-Time Skeleton-Tracking-Based Human Action Recognition Using Kinect Data, Georgios Th. Papadopoulos et al. 2014

Due to the requested revisions by the reviewer, we already altered the erroneous, initial statements regarding the offline application of pose estimation networks (please see the comments to query #1). Moreover, we toned down the initial sentences in the discussion claiming the limitation of current solutions to offline analysis.

In the “Discussion” of the revised manuscript (Page 5, line 39-40), the sentence reads as follows:

“Here we take advantage of the power of DLC’s offline body part tracking to train a neural network and integrate it into our real-time, closed-loop solution.”

15. “Speed” is mentioned as an example trigger, yet, the system as it is presented, only supports single-frame postural information as triggers. In general, it should be elaborated how sequences of actions / postures are to be treated and processed, as well as accompanying implications for changes in processing times, when the application of filters become important.

We agree with the reviewer that the initial manuscript would benefit from additional information on the use of action/posture sequences and resulting performance changes for DLStream.

To clarify, DLStream is running without batch processing multiple images, but with frame-by-frame analysis via DLC trained networks. Importantly, any number of extracted pose estimations can be stored within DLStream (e.g. in the experiment or trigger module) and be used for posture sequence-based triggers.

In the current version of the manuscript, we did not actively show an experiment with such a trigger as both reported use cases can be sufficiently solved by using single-posture triggers. To give potential users additional information on the capabilities of DLStream, we recently released a speed-dependent trigger together with extensive tutorials/guidelines on our GitHub page (<https://github.com/SchwarzNeuroconLab/DeepLabStream/wiki/>). As a proof of concept for a functional speed trigger, we added a Supplementary Video (Suppl. Video 1) in the revised version of the manuscript.

16. The distinction between “task” and “activity” is somewhat vague. Clear definitions for both need to be provided.

In the revised version of the manuscript, we now keep track of clear definitions and a clear distinction of the words “task” and “activity”.

The sentence now reads (Page 1 Line 11-13):

“Optogenetic manipulation, for example, offers high temporal precision, enabling the manipulation of experience during experimental tasks that test mechanisms of learning and memory¹²⁻¹⁴, perception^{15,16} and motor control^{17,18}.”

The sentence regarding “activity” now reads (Page 7 Line 13-15):

“With DLStream, we show that the real-time detection of specific behaviors in freely moving mice can be combined with neuronal activity-dependent labeling systems (Cal-Light) to investigate the neuronal correlates of behavior.”

17. As actions can seldomly be encapsulated by a single frame, action recognition should be coupled with the extracted posture. The authors need to elaborate on how action recognition would be implemented in their approach and where potential difficulties lie.

We thank the reviewer for this remark and included changes in the “Discussion” of the revised manuscript that now discuss the challenges and difficulties of manual definition of action-recognition and included a prospect to machine-learning based approaches that are already available to the field and should increase the availability of triggers to the community (Page 6 Line 1-17). We also referred to the tutorial and guides section to help users learn how to modify DLStream (<https://github.com/SchwarzNeuroconLab/DeepLabStream/wiki/>).

18. Supplement fig 3 does not seem to be available! (referring to bottlenecks in processing speeds)

We apologize for the inconvenience. The supplemental information is now fully available.

19. Without knowing more about the implementation of DLStream, the comparison to similar work regarding framerate but not response time without hardware specifications gives little insight into the comparability between the two systems (Forys et al. 2020). As mentioned in (13.), these details are crucial.

We now included a supplementary table that includes performance tests for several different configurations and compared them in the discussion to the already existing publications (Mathis et al. 2018, Kane et al. 2020, Forys et al. 2020). Note, that the toolkit includes a function to benchmark the user’s system (Page 5 Line 26-34, Page 7 Line 35-38; Supplementary Table 1, please also see the answer to query #2).

Issues regarding the hardware latency are also addressed (please see the answer to query 2, or Page 5 Line 12-16, Page 7 Line 20-23 in the revised manuscript)

20. Reference to additional works regarding multiple animal tracking requires citation.

We included the necessary citation for sLEAP (Pereira et al. 2020) and id-tracker (Romero-Ferrero et al. 2019) to give the reader an idea of the multiple applications that can track multiple animals. At the time of submission, our implementations of pose estimation networks was limited to DLC based approaches, therefore we mainly discussed maDLC. In the revised version of the manuscript, we now include experimental implementations of other pose estimation sources (Leap (Pereira et al. 2019), DeepPoseKit (Graving et al. 2019)) increasing the spectrum of available options to the potential user. We also added a short paragraph in the “Discussion” to explain our current approach to multiple animal tracking (Page 7 Line 43-50, Page 8 Line 2-5).

• Code availability:

21. At the moment, the link is not accessible, therefore it is not possible to judge the claims made about the versatility or limitations of the created software. Please provide your written code for the continuation of the revision process.

We apologize for the inconvenience and fixed the error (<https://github.com/SchwarzNeuroconLab/DeepLabStream>).

• Methods:

22. (Conditioning task) Please provide a number for the number of animals that correctly learned the association in the negative trial.

None of the tested mice reached the hypothetical 85% criterion in the negative trials (mice not going to the reward location during visual stimulation), the reason being that the aversive stimulation is presented regardless of the position of the mouse in the arena. Thus, the mice are not conditioned to associate their position with the negative stimulus.

To be clearer on this issue we added the following sentence in the methods part of the revised manuscript (Page 13, Line 46-47):

“We did not evaluate the success rate of negative trials since the aversive stimulation was delivered regardless of the animal’s behavior.”

23. Head direction-dependent optogenetic stimulation and action dependent light stimulation share a lot of information. Consider combining the two sections

We thank the reviewer for this comment and combined both sections. We included the necessary information from the section “action dependent light stimulation” in the section “head direction-dependent optogenetic stimulation” in the “Methods” section of the revised manuscript (Page 14 Line 11-30).

24. (Experimental Setup) If the depth data was not used, why were higher framerates not considered? With the argument that pushing the framerate further to test the upper limits of the online DLC network, this would have made a better comparison to Forsys, et al. possible.

For clarification: Initially we planned to also incorporate depth data in our experiments and developed DLStream accordingly. However, we found that with our current setup, depth-extraction was not reliable enough (different camera performance levels) for our purposes.

In comparison to Forsys et al. 2020 and the recently released preprint from DLC-live (Kane et al. 2020), we now incorporated new performance tests with multiple camera configurations to address this issue (see Supplementary Table 1, please also see beneath query #2). We show that DLStream’s performance is in line with the reported inference speed of DLC-live networks when comparing similar hardware resources (GPU) and input size (frame resolution). We are addressing this point by including a new section in the “Results” part of the revised manuscript (Page 5 Line 26-34).

25. Has there been any testing regarding fewer or more tracked body parts? An increase in network complexity for a larger number of score maps might influence inference time significantly.

We thank the reviewer for this remark. Accordingly, we tested different network architectures (ResNet50, ResNet101 and MobileNetV2) as well as more body parts (9 and 13) and included the new information in the additional Supplementary Table 1 in the revised manuscript.

Indeed, the choice of architecture in combination with the number of tracked body parts and the input frame resolution reduces the available inference speed of DLStream at higher performance levels (> 60 fps / 320x256 px resolution). Details are given in the revised manuscript (Supplementary Table 1 and “Results” section, Page 15, Line 36-51, also beneath query #2).

26. It is unclear, whether modules of “possible body parts” need to be custom built for comparison of current states of the animal. It is important to elaborate on how a user of the developed system would introduce new cues / modules.

To clarify; DLStream is currently using the config files directly from DLC networks which enable the identification of user estimated body parts by name when designing an experiment. Any number of previously defined body parts can be incorporated into newly designed modules and already provided triggers can be easily adapted to fit the user-defined naming system used in DLC-model training. To support accessibility for DLStream, we currently released extensive tutorials/guides on the GitHub page to advise potential users how to design their own triggers or adapt existing ones to their needs (<https://github.com/SchwarzNeuroconLab/DeepLabStream>). Additionally, we added a section in the “Methods” part of the revised manuscript addressing these issues (Page 16 Line 13-29).

Please note that we added an entirely new section in the “Methods” part of the revised manuscript “DLStream output and adaptability” and summarizing the core aspects of currently released tutorials

(<https://github.com/SchwarzNeuroconLab/DeepLabStream/wiki/>) since also reviewer #3 requested this information.

• Minor language corrections:

27. P11, "The screen was the same size as the arena wall it was placed at."

corrected (Page 13 Line 49).

28. P11, "Directly after the surgery animals were administered 1 ml 5% Glucosteril solution"

corrected (Page 13 Line 27).

Reviewer #2 (Remarks to the Author):

The manuscript under review by Schweihoff et al titled “DeepLabStream: Closing the loop using deep learning-based markerless, real-time posture detection” provides an open-source platform for autonomous real-time pose-estimation (via DeepLabCut) and close-loop behavioral experimentation. Specifically, the authors develop and provide a Python-based pipeline that is validated using two behavioral procedures, in (1) a second-order olfactory conditioning task and (2) an opto-tagging head direction task.

Comments:

1) DeepLabCut is a very popular, and highly used, pose-estimation platform. However, there are numerous others (and surely more to be developed in the near future), and it would strengthen the use and general applicability of this pipeline if it were validated with other pose-estimation architectures like DeepPoseKit, LEAP (or SLEAP) and also the more recent maDLC release. At the least, a discussion on how DeepLabStream (abbreviated DLS from here) can, or cannot, be used with these other architectures should be included. If there are technical reasons DLS will only work with DLC, this should be explicitly explained.

Currently, DLStream (Release version 1.21) is not able to utilize DeepPoseKit, Leap/sLeap networks or multiple animal networks trained with maDLC. The technical reason for this is now discussed explicitly in the “Discussion” section of the revised manuscript (Page 7 Line 39-50). Please note that we are actively developing an extension of the currently available network frameworks and released an experimental implementation of DeepPoseKit exported models (LEAP, StackedDenseNet, StackedHourNet, DLC) as well as maDLC and DLC-Live exported DLC models illustrating the technical feasibility for implementation in our GitHub repository (<https://github.com/SchwarzNeuroconLab/DeepLabStream>). So far, DLStream is compatible with any DLC-trained network (including multiple animal networks trained outside of maDLC).

2) Is DLS only compatible with single-animal experiments? Can multiple (dissimilar) looking animals be pose-estimated concurrently, or similar looking animals with maDLC? I imagine the addition of more pose-estimation features may lead to latency. How many features can be pose-estimated on a single, or multiple, animals before latency becomes too challenging to work with?

The question of the reviewer is very valid. Indeed, we build DLStream to actively support multiple animal pose estimation and even included our own simple multiple animal pose estimation solution, which we discontinued and instead adapted our code to current DLC development stages. As mentioned above in query 1, we are already working on an update that incorporates the necessary code changes to include the new pose estimation for multiple animal networks in maDLC (<https://github.com/SchwarzNeuroconLab/DeepLabStream/tree/dev>). Please also note that currently multiple animal tracking is possible if the user is tracking dissimilar looking animals. These networks are using the original pose estimation from DLC version $\leq 2.0.0$ and tracked body parts can therefore be utilized in DLStream without the necessity to assign identity.

In line with requests from reviewers #2 and #3, we included a performance test for multiple network architectures (ResNet50, ResNet101, MobileNetV2), image resolutions and number of body parts (new Supplementary Figure 3), which suggest that pose estimation of two animals would result in a reasonable latency when the right combination of image resolution, network architecture and hardware configuration are chosen (e.g. 9 body parts for 3 animals with a ResNet50 model and 640x512 px frames would result in a framerate of ca. 40 fps). We added a section to the “Discussion” part of the revised manuscript to address this comment (Page 7 Line 35-38).

3) More information on the table-based file (mentioned on page 2) would be useful, as understanding the data frame can often be the crux of adopting and adapting an open-source package. This deserves at least some mention in the Methods.

Also, in line with requests from reviewer #1 and #3, we addressed this request in the revised manuscript by adding a new section in the “Methods” part as suggested (Page 16 Line 13-30). We also released extensive tutorials/guides on the GitHub page to teach users how to work with DLStreams output (<https://github.com/SchwarzNeuroconLab/DeepLabStream/wiki/>).

The corresponding wiki section reads as follows:

„The understanding of the output of any software is crucial for the adoption and adaptation, especially in the open source community. Even if authors are thinking that their software's output is easy to understand, it never hurts to explain it again. In the following section, we will go over the major layout of DLStream's output step by step, so that you can start analysing your experimental results right away. Let's start with the general output. DeepLabStream is outputting a raw video and a table-based csv file for every dlc-enabled session that you run with it. While the video is a simple recording of the entire session (if you pressed "Start recording" or started an experiment) and quite straight forward, the table-based file can be harder to understand and work with. You can open the csv file with any notepad or excel (or directly in python).

Here is a modified example from one of the experiments used for the paper:

```
;Animal1;Animal1;Animal1;Animal1;Animal1;Animal1;Experiment;Experiment;Time  
  
;neck;neck;nose;nose;tailroot;tailroot;Status;Trial;  
  
;x;y;x;y;x;y;;  
  
Frame;;;;;;;;;  
  
1;464.0;131.0;475.0;127.0;427.0;131.0;False;False;0.001  
2;423.0;336.0;409.0;345.0;436.0;318.0;False;False;0.033  
3;425.0;336.0;412.0;346.0;437.0;318.0;True;False;0.066  
4;425.0;336.0;412.0;346.0;437.0;318.0;True;False;0.1  
5;424.0;336.0;411.0;348.0;437.0;317.0;True;True;0.133  
6;424.0;336.0;411.0;348.0;437.0;317.0;True;True;0.166  
7;424.0;336.0;414.0;348.0;437.0;317.0;True;False;0.199  
8;424.0;336.0;414.0;348.0;437.0;317.0;True;False;0.243
```

You can see that the general structure is made up by entries separated by ";" (a delimiter). When importing the file to excel for example, you need to specify the delimiter in order for excel to detect new columns/cells. In addition you can see that the headers are actually multiple rows, detailing the underlying numbers. We start from left to right and from top to bottom. As you have probably already guessed, "Frame" is referring to the discrete time series (the frame sequence) and acts therefore as an index as well. "Animal1" stands for the tracked animal and any pose estimation data (in multiple animal experiments the second animal would be "Animal2" and so on), while "Experiment" is a collection of experiment based parameters that can change frame-by-frame. The entry "Time" however is measuring the time that has passed since the start of the experiment and gives a good estimation of your overall performance (e.g. 0.032 ms between Frame 1 and 2), the performance time is the time between two full loops in DLStream (including the whole experiment) and its main time demanding component is the prediction itself. Coming to the next lines we see the DLC-style reporting of bodyparts (neck, nose, tailroot) and their estimated x/y positions. "Status" is a boolean parameter that states whether the experiment was started using the "Start Experiment" button. The unprocessed csv file includes all pose estimation data from the moment of "Start Analysis", which has to be initiated first in order for the network to be at "full speed" before starting any experiment. If you want to restrict your output to the experiment, you can use this column to filter for TRUE values. The "Trial" column refers to the experiment specific output. It can be a bool (as in the example) or a string that specifies which trial was active (e.g. in the conditioning task the trial column would specify if it was an aversive or appetitive stimulation trial). Look at the experiment section to learn more about this output. If you are using a simple stimulation (e.g. optogenetic) a boolean is the best way to ease any future analysis.

If you are familiar with python programming and pandas, I recommend using the following lines for easy import of DLStream's csv output:

```
import pandas as pd
```

```
df = pd.read_csv('path_to_csv',header=[0,1,2],sep=';',index_col=0)
```

Sometimes `pd.read_csv` will not read empty rows correctly (due to an issue with multiple indexing and unique names). One way to solve this is to remove the automatic naming again with this (See <https://stackoverflow.com/questions/41221079/rename-multiindex-columns-in-pandas>):

```
def remove_unnamed_lvl_multiindex(df: pd.DataFrame):
```

```
    for i,columns_old in enumerate(df.columns.levels):
```

```
        columns_new = np.where(columns_old.str.contains('Unnamed'),",",columns_old)
```

```
        df.rename(columns=dict(zip(columns_old,columns_new)),level=i,inplace=True)
```

```
    return df
```

In the future, we plan to release a few utility scripts that will help you to process results obtained from DeepLabStream. Until then I hope that this short introduction proves useful!"

4) 30FPS is adequate for many, but not all, behaviors as noted in the discussion section. The bottleneck is likely determined by the resolution of the captured video. A parametric dataset using resolution downsampled comparisons to the original 848x480 video would be very helpful in understanding how DLS could be adapted to behaviors that require higher sampling. Notably, a lot of behaviors are recorded at lower resolutions, and DLC can pose-estimate lower resolution images just fine. Would be useful information for users!

We thank the reviewer for this comment. Again in line with requests from the other reviewers we tested a different camera (Basler acA1300 – 200 um) and we included a performance test for multiple network architectures (ResNet50, ResNet101, MobileNetV2), image resolutions and number of body parts (new Supplementary Figure 3). We also added the results to several sections of the “results”, “methods” and “discussion” parts of the revised manuscript (Page 5 Line 26-34, Page 7 Line 35-38, Page 15 Line 43-51).

5) Minor comment, but the github link provided in Code Availability on page 7 leads to a 404 due to the hyphen.

We apologize for the inconvenience. Unfortunately, our formatting added a line-break (with a hyphen) in the middle of the link, which we did not notice. The link will work in the final release ((<https://github.com/SchwarzNeuroconLab/DeepLabStream>)).

Overall, I think this protocol will be useful to the field and is developed to a point that is suitable for publication.

Many thanks for this overall positive evaluation of our manuscript.

Reviewer #3 (Remarks to the Author):

Schweihoff et al. present DeepLabStream, a system for real-time tracking and stimulus delivery using DeepLabCut, and demonstrate its performance in two example experiments. Such a tool has many potential applications in systems neuroscience, and the authors' efforts to create and share such a tool with the community is laudable. However, the impact of the paper is somewhat reduced by the two example experiments provided, where issues of experimental design make it hard to know how well DLStream is actually working. A simpler example experiment (e.g. classical conditioning or conditioned place aversion) would have made a much more convincing case that the system works as advertised. In addition, a more rigorous evaluation of DLStream's performance is needed. More specifically, I have the following comments:

The evaluation of DLStream's performance is the most important part of the paper, but it is currently a bit limited. A few points on this topic:

1) The authors characterize the accuracy of DLStream by computing the average Euclidean distance between nose, neck, or tail points on consecutive frames, as well as the average head direction variance between frames. As the authors themselves note, these values depend on two terms-- the actual movement of the animal during the experiment, and the performance errors of the pose estimate. Thus there is no way for the reader to determine from the reported Euclidean distances/head directions how accurate the system's pose estimations really are. Because of this, I believe that it is necessary for the authors to collect a novel, manually annotated "test set" of nose/head/tail point locations, and to evaluate DLStream accuracy with respect to this test set.

We thank the reviewer for this suggestion. We agree that a more thorough report on the network performance would be beneficial for readers to evaluate the successful implementation of DLC trained networks in our conducted experiments. Therefore, as requested by the reviewer, we generated the proposed, manually annotated data set (n = 597 labeled frames) and estimated the pose estimation error per body part, as well as the error in the extracted head direction angle. The results are added to the corresponding "Results" section as well as the "Methods" section in the revised manuscript (Page 4-5 Line 38-2, Page 7 Line 35-38, Page 16 Line 4-12).

Revised "Results" part:

"Note that, due to the inherent individual network performances, DLStream's effective accuracy in posture detection is heavily influenced by the previous training of utilized networks. Nevertheless, if performance is not sufficient for the executed experiment, DLC networks can always be retrained using the DLC provided tools. In our hands, the trained network used during optogenetic experiments resulted in an estimated average pose estimation error of 4 ± 12 pixels (px) for the neck point, 3.3 ± 4.4 px for nose and 3.3 ± 2.0 px for tail root (n = 597) when compared to a human annotator labeling the same data set (mice without tail were ~60 px long in our 848x480 px recordings). Body part estimation resulted in an average head direction variance of $3.6 \pm 9.6^\circ$ (tested in 80 sessions for 1000 frames per session) between consecutive frames with an estimated average error of $7.7 \pm 15.1^\circ$ compared to human annotation (n = 597) per frame. The frame-by-frame variance is a product of performance errors and the inhomogeneous movement of the animal during experiments while the difference between network pose estimation and human annotation is most likely a result of inaccurate tracking which can be reduced by additional training and/or bigger training sets. Note that depending on the mixture of episodes of fast movements and slow movements during sessions, the variance might change."

Revised "Methods" part:

"To evaluate our own DLC trained networks, we measured the pose estimation error and compared it to a human labeled dataset. For this, we extracted a new image set from our optogenetic experiment sessions (n = 597) and measured the average difference (Euclidean distance) between human annotation and pose estimation in position, as well as resulting head direction angle. Additionally, we calculated the false positive/false negative rate of hypothetical head direction triggers with differently sized angle windows (60, 50, 40, 30, 20, 10). To counter any non-uniform distribution of head direction angles, we averaged the rates for multiple ranges per bin (e.g. 0-60°, 60-120°, 120-180°) and calculated the standard deviation. See Supplement Fig. 3 for details. "

2) While the *average* error of pose estimation may be low, what is more important is how often DLStream produces false positive/false negative event detections. This could be computed by, for

example, having a human who is blind to DLStream output manually annotate when the mouse is performing the target behavior (eg looking at the screen or smelling a petri dish in the conditioning experiment, or facing the target direction in the Cal-Light experiment), and comparing the accuracy of DLStream to these manual annotations. While time consuming to collect, it is critical that these performance measures are established, or the reader will be left with no way to evaluate how usable DLStream might be for their own work.

We thank the reviewer for suggesting an additional measure to better allow readers to evaluate the performance of our trained networks and better interpret the outcome of the experiments. As suggested by the reviewer (query 1), we used the annotated data set (n = 597) to evaluate the event detection accuracy (false positive and false negative rate) of our experimental head direction trigger. Additionally, an experimenter blind to the DLStream output manually annotated all event detections (false positive and positive) occurring in three videos of complete sessions from different mice (n = 164, videos were randomly chosen). We then calculated the false positive and false negative rate for a number of differently sized angle windows (60, 50, 40, 30, 20, 10; In line with query 3). Specifically, we counted events in which the human estimated head direction angle was within the detection window and compared it with the network detection. The relevant false positive event detection rate was 11.6 ± 4.8 % in the annotated data set and 11.8% for the evaluated sessions/videos.

The exact results are presented in the “Results” and “Methods” section, as well as in a new supplementary figure in the revised manuscript (new Supplementary Fig.3, Page 3; Page 5 Line 2-8 Page 16 Line 4-12, please also see the answer to query #3).

3) It would also be helpful to see how the above performance depends on the trigger used: for example, in the Cal-Light task, what is the false positive/negative rate as a function of the stimulation angle around the target direction? (e.g. at 60 degrees, 50, 40, 30, 20, 10, 5...)

In line with the requested data from the reviewer in query 2, we used the manually annotated data set to estimate the pose estimation error rate (false positive and false negative) for different trigger window sizes (60, 50, 40, 30, 20, 10). To counter a non-uniform angle distribution, we used multiple windows of the same size ranging the full 360° (0-60°, 60-120° etc.) and averaged the resulting detection rates. Due to the pose estimation error which excludes head direction angles $\leq 10^\circ$ (for description of the calculation please see query 1), data on the detection rates for windows smaller than 10° was not included.

The exact results are presented in the new supplementary figure (Supplementary Fig.3, page 3).

For the reviewer’s consideration, we included the new supplementary figure below:

b

Window size [°]	False positive detection [%]	False negative detection [%]
60	11.6 ± 4.8	11.1 ± 4.1
50	13.2 ± 5.3	13.7 ± 5.4
40	14.7 ± 5.8	14.7 ± 5.9
30	20.1 ± 11.2	19.8 ± 10.5
20	29.0 ± 20.7	28.2 ± 18.7
10	72.8 ± 75.6	75.8 ± 89.0

“New Supplement Fig. 3 | Estimation of accuracy of head direction triggers with different angle window sizes.

a, Histograms (10° bins between 0-360°) of the distribution of the labeled dataset (n = 597), with human annotation (right) and head direction angle based on network pose estimation (right) using the network trained for the optogenetic stimulation task. **b**, Table showing the false positive and false negative detection rate of the network pose estimation against human annotation in several differently sized angle windows (theoretical triggers). To counter any effects of non-uniform distribution, the window was moved around in steps and the average, as well as the standard deviation was taken from all detected events. An event was counted as false positive if the pose estimation resulted in a head direction within the window, while the human annotation did not (and vice versa for false negative).”

4) The paper does not explain how the end user constructs triggers- is there an API or user interface? The paper needs a section to explain, step-by-step, how triggers are actually created by an end user of the system, and what conditions can be used to create a trigger. This is ostensibly done in the Discussion, however only in broad terms, with no references to how these actions are supported by the DLStream software/interface.

To facilitate the accessibility of DLStream, as very reasonably pointed out by the reviewer, we now added a section in the “Methods” part of the revised manuscript that explains the adaptation/design of triggers in detail (Page 16 Line 13-29). We also extensively worked on our GitHub page that now contains several additional tutorials (<https://github.com/SchwarzNeuroconLab/DeepLabStream/wiki>). The tutorials on our GitHub page are meant as a valuable extension to the revised manuscript and are updated in a timely fashion to continuously support the open-source character of DLStream.

5) The Discussion also makes contradictory statements about what kinds of triggers an end user can build. One sentence states “Out of the box, DLStream supports triggers based on single-frame postural information, although posture sequences or complex behavior modules are also possible once behavior based on collected posture data has been classified, modeled, and integrated as trigger modules into DLStream.” This makes it sound like a potential end-user of DLStream would only have access to single-frame posture triggers, however earlier in the paragraph the authors state “With DLStream, a combination of triggers based on the animal’s posture or posture sequence is now possible.”

To avoid confusion, we changed the corresponding section in the “Discussion” part of the revised manuscript (Page 5-6 Line 48-3).

The sentences now read:

“With DLStream, a combination of triggers based on the animal’s posture or a sequence of postures can be integrated into experimental designs. Example triggers include center-of-mass position, direction, and speed of an animal, although multiple individual tracking points can also be utilized, such as the position and trajectory of multiple, user-defined body parts. This allows the design of advanced triggers that include head direction, kinematic parameters, and even specific behavior motifs (e.g. rearing, grooming or sniffing). Out of the box, DLStream supports triggers based on single-frame as well as sequential postural information, although complex behavior modules could also be utilized once behavior based on collected posture data has been classified, modeled, and integrated as custom trigger modules into DLStream.”

Please also see answer to query 15 of referee #1.

In general, the authors should tone down the “sales pitch” style language in their introduction and discussion and state the functionality and limitations of their system directly.

In the revised version of the manuscript we tried to use a language that clearly states the functionality and limitations of our system (Abstract, Page 5 Line 48-49, Page 7 Line 19-21, Line 42-50, Page 8 Line 6).

We regret that the reviewer got the impression of a “sales pitch” style language. From the beginning on our intention was to present a well-structured toolkit that can be used to create custom experimental solutions. To support this notion, we significantly improved the documentation and increased the accessibility for potential users (<https://github.com/SchwarzNeuroconLab/DeepLabStream/wiki>).

The revised abstract now reads:

“In general, animal behavior can be described as the neuronal-driven sequence of reoccurring postures through time. Most of the available current technologies enable offline pose estimation with high spatio-temporal resolution. To correlate behavior with neuronal activity it is often necessary to detect and react to behavioral expressions in real-time. Here we present DeepLabStream, a versatile closed-loop tool providing real-time pose estimation to deliver posture dependent stimulations. DeepLabStream has a temporal resolution in the millisecond range, can utilize different input, as well as output devices and can be tailored to multiple experimental designs. We employ DeepLabStream to semi-autonomously run a second-order olfactory conditioning task with freely moving mice and optogenetically label neuronal ensembles active during specific head directions.”

6) I am surprised that only 300 frames were used to train DLC: while DLC advertises that it can be trained using a small number of examples, my experience in talking to labs that use it is that in practice many more (a few thousand) frames are needed to achieve satisfactory performance. Have the authors confirmed that DLC performance has plateaued with only 300 training examples? Might more training data further improve performance of the system?

We thank the reviewer to bringing up a very valid point of implementing DLC based models. We completely agree that DLC pose estimation is heavily dependent on a successful training and previous selection of a representative training set. From our own experience we know that especially multiple animal tracking requires a huge amount of annotated training examples. However, within our implementations (single animal, high contrast) the small number of training data yielded satisfactory results when models were trained to a plateau with 500k iterations (please also see the answer to query #1). This is also in line with the recommendations previously reported by Mathis et al. 2018.

7) It is unclear from the text whether DLStream is compatible with other choices of tracked body parts (instead of the nose-neck-tail points tracked here.) Is this the case? If so, how is the response time of DLStream affected by the number of body parts tracked?

In the revised version of the manuscript we now clearly stated that it is possible to apply user-defined body parts as input (Page 5 Line 51, new Supplementary Table 1).

Please also see the answer to query #25-26 from reviewer #1.

I also had the following comments on the two experimental use-cases presented:

8) The results of the second-order conditioning task are a little unusual. As noted in the text, the strongest effect present in Fig 3E seems to be that the mice are spending more time in the quarter of the arena where they previously received a reward. However, I don't see evidence for a "clear odor preference" as claimed in the discussion: none of the plots show that the mice investigate the Vanillin more than either of the two novel odors, nor do the mice spend less time investigating the Rose dish than the novel odors. The odor location plots do show a statistically significant difference between Vanillin and Rose investigation times, but the confound of the reward port location makes the plots hard to digest. The fact that the two groups show very different investigation times towards Acetophenone also makes me distrust the odor preference results.

We agree that the initial setup of the experiment made it hard to conclusively interpret the results of the second order conditioning task. We therefor decided to repeat the entire experiment in line with the suggestion by the reviewer in query #9.

9) While there's a lot going on in this experiment (re odor location/reward location interactions), there's little in the text that attempts to unpack these complexities of the data and justify the claim of a learned odor preference-- the authors instead focus on their finding of a preference for the reward corner. While this is a valid finding, it is problematic for this paper because preference for the reward corner is a behavior that would emerge whether or not the DLStream system was actually working. Would it be possible to repeat this experiment with the reward port in a location equidistant to the two odor sources (eg at the center back of the arena) so that the performance of DLStream could be evaluated without having to worry about this added confound of reward location?

We repeated the experiment and included the new results in the "Results", "Methods" and "Discussion" section of the revised manuscript, as well as in the revised version of Fig. 3 (Page 3, Line 21-22 + 35-43, Page 6 Lin 51-53, Page 13 Line 46-47, Page 14 Line 5-10). Thanks to the suggestions from the reviewer, we were able to eliminate the place preference issue by moving the reward port location to the middle of the arena wall and additionally exchanged the arena that was used for the odor preference task (please see revised Fig. 3). To further support interpretability of our data, we altered the bar plot of the new group to include data points and altered/simplified the figure to clear up some of the complexity issue of the presentation of this experiment (revised Fig. 3 a, c, d).

For the reviewer's consideration, we included the revised figure below:

“Revised Fig. 3 | Closed-loop conditioning task. a, Conditioning. When a trial is triggered by the mouse facing the screen (left, green triangle and ring), the mouse is shown a visual stimulus (yellow lightning bolt). Mice not facing the screen do not receive the stimulus (red x). In the positive trial (middle; green lightning bolt, green line), a reward is delivered (blue drop, arrow down) and withdrawn (blue drop, arrow up) if not collected within a preset time period. In the negative trial (right; blue lightning bolt, blue line) only a loud tone (red polygon) is delivered. **b, 2nd Order Conditioning.** Upon exploration of either odor location (colored black circle) the mouse is shown one of the previously conditioned visual stimuli on the screen (left, yellow lightning bolt). Conditioning was conducted in two stages (top middle and top right). The first stage (top middle) consisted of direct contact with the odor location, while the second (top right) was dependent on the proximity of the mouse to one of the locations (black arrow) and the mouse facing towards it. **c, Odor Preference Task.** The mouse was set in an open field arena with one odor in each of the quarters (colored circles). The total investigation time of each odor source was measured. **d, Investigation time during odor preference task in odor location: ROIs encircling the odor location.** Bar graph shows STD and individual data points. $p < 0.05$ (*) paired t-test; $n = 6$ (2 trials per mouse, 1 trial excluded). V = Vanillin (S+), R = Rose (S-), VA = Valeric acid, A = Acetophenone”

10) There are several design decisions in the Cal-Label experiment that seem strange. In Figure 4c, the red vs gray portions of the right panel seem to suggest that light stimulation was only actually delivered for a tiny fraction of the time that the tracked mouse’s head was pointed in the target direction (blue arc). Is this all because of the 15-second inter-stimulus time? This extreme delay is not commented on in the main text, however the methods cite that this is “to avoid overheating of brain tissue”. This seems strange- other groups have delivered light stimulation for much more than 5 seconds without issue. If overheating is truly a concern, why not reduce the power, stimulation frequency, or duty cycle of the laser? Or if all this is not possible- why not narrow the target direction to less than 60 degrees?

The experiment was designed such that only a fraction of potential stimulation events was used. We followed the recommendations originally published in Lee et al. 2017, however, reduced to inter stimulus time to 15 seconds (from 25 sec) since this duration to our experience is more than sufficient to avoid tissue heating. Consequently, a session was 30 minutes long, anticipating a total stimulation time per session of about 100 seconds to match the total stimulation time necessary to achieve light-induced labelling by the Cal-Light system (Lee et al. 2017). Additionally, we observed in preliminary experiments that mice moved quickly through several degrees of head direction, with periods ranging from sub-second to multiple second episodes in the same target angle. Therefore, we restricted light stimulation events to durations ranging between 1 and 5 seconds.

11) It also seems like light stimulation is preferentially delivered when the animal's head is between ~-30 (ie 330) and 0 degrees, with much less stimulation between 0 and 30 degrees. Why this difference?

While the overall, resulting stimulation was within the designed window in all individuals, animals showed variation in head direction and preferred location. In addition, we now included different examples demonstrating the variability in a new supplementary figure (new Supplementary Fig. 4; Page 5).

For the reviewer's consideration, we also included the new supplementary figure below:

“New Supplement Fig. 4 | Examples of head direction angles during optogenetic light stimulation.

a-b, Left: Example radial histogram of all head directions (5° bins) during stimulation (red) within one session (normalized to the maximum value). Right: Radial histogram of all head directions during the whole session (grey) and during stimulation (red) (normalized to the maximum value of the entire session). Rings represent quantiles in 20 % steps. Each panel shows a session from a different mouse. **c**, Example radial histogram of all head directions (same representation as in **a-b**) from the same mouse shown in **a** in the next session. Note that the mouse is showing different distributions of head direction between sessions in both the stimulation events and the overall session, while the stimulation is mostly limited to the angle window (thick blue arc)”

12) In figure 4e, the legend claims this is a heatmap of “the relative occupancy of the mouse within the arena during the session”, however text claims 4e shows occupancy only during times when stimulation was delivered. Which is it? Ideally you should show both occupancy during the session and occupancy during stimulation, as well as the ratio of the two (ie the conditional probability of stimulation given occupancy). Also, the text states “light stimulation was independent of the animal's position in the arena”—what is the basis of this claim? 4e shows a pretty dense occupancy peak at the arena wall, just to the left of the stimulation zone (dark red), suggesting that the animal is in fact preferentially stimulated when it is in this location. This could occur if the animal spends most of the time circling the edges of the arena in a fixed direction, rather than exploring randomly. Perhaps placing a few small food rewards throughout the arena could encourage more varied behavior.

Unfortunately, we could not retrace the text passage claiming that Fig. 4e shows the occupancy only during times when stimulation was delivered. In the main text we say that: “Mice explored the complete arena during the task and the resulting light stimulation was independent of the animal's position in the arena (Fig. 4e).” Our claim remains that the figure shows “the relative occupancy of the mouse within the arena during the session”. To clarify this issue, we now in the revised version of the manuscript show an additional heatmap illustrating the relative occupancy during times when stimulation was delivered as suggested by the reviewer (revised Fig. 4 f, page 20-21).

We also rephrased the sentence claiming that light stimulation was independent of the animals position in the arena, to “Mice explored the entire arena during the task and the resulting light stimulation was not dependent on the animal’s position in the arena, as animals could angle their head in the target orientation from any position within the arena (Fig. 4e-f).”

In order to encourage a more varied behavior in this type of arena we did exactly what the reviewer suggested. We placed small food rewards at random locations inside the arena during arena habituation as mentioned in the “Methods” section of the manuscript (Page 14, Line 14-16). We did not place the food rewards during the stimulation sessions to keep the experimental conditions as constant as possible.

For the reviewer’s consideration, we included the revised figure below:

“Revised Fig. 4 | Optogenetic labeling of head direction-dependent neuronal activity. a, Left: Stereotactic delivery of Cal-Light viruses into the ADN and fiber ferrule placement. Middle: Infected neurons (red) are stimulated with blue light (488 nm) controlled by DLStream. Right: infected neurons are only labelled (yellow) when they are active (black waves) during light stimulation (middle). **b,** Schematic drawing of the circular arena with the visual cue (thick black arc) and the target window (thick blue arc) around the reference point (red circle). DLStream triggered stimulation is strictly dependent on the correct head direction (blue arc). **c, Left:** Representative example radial histogram of all head directions during stimulation (red) within one session (normalized to the maximum value). Mean resultant vector length is indicated by r . Right: Radial histogram of all head directions during the whole session (grey) and during stimulation in 20% steps. Rings represent quantiles in 20% steps. **d, Left:** Representative random sample of the whole session simulating stimulation without DLStream control at random time points during the session (normalized to the maximum value). Mean resultant vector length is indicated by r . For each session, random distributions were calculated 1000 times. Right: For one session, distribution of mean resultant vector lengths generated by random sampling ($n = 1000$). The red line denotes the actual mean resultant vector length during stimulation in the session. The dotted black line represents the $p < 0.01$ cutoff. **e,** Representative example of the mouse’s position (grey) over time during the first 5 minutes of the session in **c**. The stimulation events are shown in blue. **f,** Heatmaps representing the relative occupancy of the mouse within the arena during the whole session (top) and stimulation (bottom) in **c**. Cue and target window are shown in their relative position. **g,** Example of Cal-Light expression in an experimental mouse. Left: tdTomato expression (red) indicating expression of Cal-Light viruses with nucleus staining (DAPI, blue). Right: Activity dependent and light induced eGFP expression (green). The white box represents the zoomed in region in **h**. The bar represents 200 μm . **h,** Close up from **g** vs. a similar region. **i,** Bar graph showing Stimulation [s] vs. Headdirection. **j,** Pie chart showing the distribution of mice: 40% eGFP+/tdTomato, 54% eGFP-/tdTomato, and n = 2 mice.”

in an animal that was not stimulated with light (No Light). Left: tdTomato expression (red). Right: Activity dependent and light induced eGFP expression (green). The bar represents 50 μm . Note that control mice show no eGFP expression. **i**, Average light stimulation during each session (40 total) corresponding to head direction (60° bins) with target window (blue) indicating the DLStream triggered stimulation onset. Paired student's t-test: $p < 0.001$. $n = 10$. **j**, Ratio between infected neurons (tdTom⁺) and activity dependent labelled neurons (eGFP⁺/tdTom⁺) in mice matching selection criteria (see Methods). $n = 2$.”

13) I do not see any data that validates that the cells labeled by Cal-Light are actually tuned for the target head direction- all that's shown in Figure 4 is evidence that some cells were labeled. This is not acceptable; the reader has no way of knowing whether the experiment worked at all. The text claims that the Yoked control mice do not have labeled cells, but no data is shown to support this claim, and this is anyway not significant evidence that head direction cells were labeled in the Experimental group.

We thank the Reviewer for making us aware of the inconsistency in our description of the experimental aim. Indeed, we cannot and should not claim that we specifically labelled head direction tuned cells within the ADN during our experiments. We made changes addressing ambiguous statements such as “head direction ensembles” in the revised manuscript (Page 3 Line 44-47, Page 7 Line 15-16). Specifically, we changed the following sentences:

“As a second example of DLStream’s applicability, we tested the possibility to optogenetically label active neurons in the anterior dorsal nucleus of thalamus (ADN) dependent on the mouse’s head direction using the neuronal activity-dependent labeling system Cal-Light¹.”

“We here delivered light stimuli to the ADN to label neural ensembles active during specific head directions.”

Our experimental aim was to label cells repetitively active during specific head-directions in a region known to contain head direction tuned cells. The yoked control was introduced to test whether similar light duration, but head-direction independent stimulation would result in sufficiently long stimulation times for Cal-Light mediated labelling (please also see the revised Supplement Fig. 1 a-c, Page 1). In line with the previously published results from Lee et al. 2019, pairing of light stimulation and randomly active cells would not have sufficient photon yields for light-induced, activity dependent labelling. We did not include the yoked control in the main text as our proof-of-concept experiment did not yield sufficient data to support a full quantification. We addressed this issue by additional explanation in the corresponding supplementary figure and hope that our clarification about the activity-dependent labelling and the head-direction dependent stimulation did address the stated issue.

For the reviewer’s consideration, we included the revised supplementary figure below (changes to the figure legend are marked in bold):

“**Revised Supplement Fig. 1 | Optogenetic labeling of head direction dependent activity in neurons:** Yoked Group. **a**, Average light stimulation in both experimental and yoked group during each session as a function of head direction (60° bins) with target window (blue wedge) indicating the DLStream triggered stimulation onset angles. Exp: $n = 10$, black bars; Yoked: $n = 8$, grey bars. Experimental and yoked groups have the same total stimulation time, but the distribution differs such that yoked group has approximately equal stimulation times across varying head direction angles. **b**, Close up (similar region of interest as shown in Fig. 4g) of representative expression in mice from the yoked group that was stimulated based on the stimulus times taken from a previous session of a paired experimental animal. Left: tdTomato expression (red) indicating expression of Cal-Light viruses. Right: Light induced eGFP expression (green). The bar represents 50 μm . **c**, Ratio between infected neurons (tdTom⁺) and activity dependent labelled neurons (eGFP⁺/tdTom⁺) in mice matching selection criteria

(see Methods) in the yoked group. $n = 2$. Light stimulation of the same duration as in the experimental group but not the same head direction specificity was not enough to reliably activate the Cal-Light labeling system, suggesting that the resulting coincidence between activity during light stimulation of the neurons was not high/often enough to result in a sufficient number of light stimulation that coincided with the neurons activity, as the head direction dependency of the stimulation was not given.”

Other minor points

1) The legend for 3C mentions a “red square” but I do not see a red square in the figure. (Also the “yellow triangle” mentioned in the legend for 3A is not shown in the figure, but I am guessing that the gray triangles are what is meant here.)

Please note that the repetition of the entire experiment resulted in significant changes to parts of figure 3 a, c, d. We removed the “red square” sentence in the figure text and altered the figure to match the triangles (Fig. 3 Page 19, please also see answer to query #9).

2) Check Figure 4 legend- some panel lettering is incorrect, and panels h/i legends are possibly incorrect/missing.

We apologize for this inaccuracy and corrected the figure legend (Fig. 4 Page 20, please also see answer to query #12).

REVIEWERS' COMMENTS:

Reviewer #1 (Remarks to the Author):

Overall, the authors have addressed my previous concerns and suggestions well. The updated manuscript is now much more coherent and with the inclusion of triggers relating to temporal changes (demonstrated by simple action recognition involving a "running" or "not running" specimen) the claim regarding dynamic triggers is now valid. An extended discussion on action recognition with the inclusion of other pose estimation frameworks as well as a guideline towards their potential implementation now makes for a well-rounded manuscript.

The only issue I currently see is with the approach to benchmarking pose estimation, regarding the split between train- and test data. However, this might be an inaccurate interpretation and only requires clarification, rather than a repetition of experiments, as the performance of the tested architectures regarding inference time, and accuracy are as expected.

Below I have listed a few corrections and suggestions.

Abstract:

The Abstract is much improved with a straightforward problem-driven approach. Now it is apparent what the authors have developed and specifically, which shortcomings of offline pose estimation solutions are addressed.

- Abstract, line 3: Ironically, in this sentence, it is not immediately apparent that the mentioned limitation lies indeed in the pose estimation being "offline".

Introduction:

Similarly, a much-improved introduction, both in structure and content.

- P1, line 20: DeepLabCut does now support online pose estimation as well. While it is not directly tied to a feedback-based system, as is your solution, it should perhaps be briefly mentioned

- P2, line 1: the meaning of body part precision is a little ambiguous. Either specify what is meant in this context or leave it out, given "track the postures" already implies extracting information of the relation between body parts

Results:

With the inclusion of latency measurements for various setups and architectures, the authors now present a sufficiently robust evaluation of the solution's capabilities.

- P4, line 39: "DLStream"

- P4, line 44-45: Are you here referring to human accuracy in labelling (agreement between observers) or the human-labelled "ground truth"?

Discussion:

No further significant edits required.

- P6, line 5: Give a few examples from literature here, regarding designed pose-based triggers in offline behavioural classification (as hinted in P6 line 11)

Methods:

- P15, line 21: "stimulation events"

- P15, line 15-51: While it is understandable to acquire as much training data as possible to increase network accuracy, a 95/5 training/test split is somewhat unusual. Unless with extensive cross-validation, leaving only 15 out of 300 images for testing does not exactly allow for a robust evaluation. As the samples collected and annotated under controlled laboratory conditions are already bound to be somewhat similar to one another, the likelihood of overfitting is high, if the reported accuracy for each tested architecture only refers to that on the small test split. Was the same set of test images used for all architectures tested?

- P16, line 5-6: Please elaborate on whether one or multiple scorers annotated the human labelled dataset and if discrepancies between observers were quantified.

Reviewer #2 (Remarks to the Author):

I am very happy with the additional data and discussion included by the authors. I feel they have gone above and beyond in responding to my comments.

Reviewer #3 (Remarks to the Author):

The authors have done an excellent job of responding to reviewer comments. The two example use cases are much more clearly presented, and the documentation of hardware latencies, network inference framerates, and error relative to human annotations are all important additions. I do not believe that any further work on the paper contents is needed.

Response to the referees:

We once again thank the three critical reviewers for the time they took to thoroughly assess our revised manuscript. In the following we address the remaining comments made by reviewer 1 in a point-by-point response (in red).

Reviewers' comments:

Reviewer #1 (Remarks to the Author):

Overall, the authors have addressed my previous concerns and suggestions well. The updated manuscript is now much more coherent and with the inclusion of triggers relating to temporal changes (demonstrated by simple action recognition involving a “running” or “not running” specimen) the claim regarding dynamic triggers is now valid. An extended discussion on action recognition with the inclusion of other pose estimation frameworks as well as a guideline towards their potential implementation now makes for a well-rounded manuscript.

The only issue I currently see is with the approach to benchmarking pose estimation, regarding the split between train- and test data. However, this might be an inaccurate interpretation and only requires clarification, rather than a repetition of experiments, as the performance of the tested architectures regarding inference time, and accuracy are as expected.

We thank the reviewer for bringing up a very valid point. We completely agree that DLC pose estimation is heavily dependent on a successful training and previous selection of a representative training set. From our own experience we know that especially multiple animal tracking requires a huge amount of annotated training examples. However, within our implementations (single animal, high contrast) the small number of training data yielded satisfactory results even when a split of 95/5 % was done. This is also in line with the recommendations previously reported by Mathis et al. 2018. As this issue also comes up in a query concerning the method section please also see our comments there.

Below I have listed a few corrections and suggestions.

Abstract:

The Abstract is much improved with a straightforward problem-driven approach. Now it is apparent what the authors have developed and specifically, which shortcomings of offline pose estimation solutions are addressed.

- Abstract, line 3: Ironically, in this sentence, it is not immediately apparent that the mentioned limitation lies indeed in the pose estimation being “offline”.

We agree with the reviewer and revised the abstract accordingly.
The revised Abstract now reads (changes in bold):

“Abstract: In general, animal behavior can be described as the neuronal-driven sequence of reoccurring postures through time. **Most of the available current technologies focus on offline pose estimation with high spatio-temporal resolution. However, to correlate behavior with neuronal activity it is often necessary to detect and react online to behavioral expressions.** Here we present DeepLabStream, a versatile closed-loop tool providing real-time pose estimation to deliver posture dependent stimulations. DeepLabStream has a temporal resolution in the millisecond range, can utilize different input, as well as output devices and can be tailored to multiple

experimental designs. We employ DeepLabStream to semi-autonomously run a second-order olfactory conditioning task with freely moving mice and optogenetically label neuronal ensembles active during specific head directions.”

Introduction:

Similarly, a much-improved introduction, both in structure and content.

- P1, line 20: DeepLabCut does now support online pose estimation as well. While it is not directly tied to a feedback-based system, as is your solution, it should perhaps be briefly mentioned

To address this issue the revised sentence in the introduction part of the manuscript now reads: “DeepLabCut (DLC)²⁰, for example, uses trained deep neural networks to track the position of user defined body parts and provides ~~offline~~ motion tracking of freely moving animals.”

We also mention the online capabilities of DLC-Live in the discussion part of the revised manuscript. Please see Page 7 Line 16-17.

The additional sentence reads as follows:

“Recent developments by DLC regarding online pose estimation reported real-time network performances for architectures used by DLC⁵².”

- P2, line 1: the meaning of body part precision is a little ambiguous. Either specify what is meant in this context or leave it out, given “track the postures” already implies extracting information of the relation between body parts

Changed as requested:

“We modified state-of-the-art pose estimation based on DLC²⁰ to be able to track the postures of mice ~~with body part precision~~ in real-time.”

Results:

With the inclusion of latency measurements for various setups and architectures, the authors now present a sufficiently robust evaluation of the solution’s capabilities.

- P4, line 39: “DLStream”

Done.

- P4, line 44-45: Are you here referring to human accuracy in labelling (agreement between observers) or the human-labelled “ground truth”?

We are referring to human-labelled “ground truth” and indicated this in the revised text (Page 4, line 15)

Discussion:

No further significant edits required.

- P6, line 5: Give a few examples from literature here, regarding designed pose-based triggers in offline behavioural classification (as hinted in P6 line 11)

Changed as requested.

“The challenge in manually designing triggers for relevant behavior is similar to the challenges faced in offline analysis, where it has already been done for a variety of relevant read-outs, **such as described in VAME²⁵, B-SOID²⁶ and SIMBA³⁸.**”

Methods:

- P15, line 21: “stimulation events”

Changed.

- P15, line 15-51: While it is understandable to acquire as much training data as possible to increase network accuracy, a 95/5 training/test split is somewhat unusual. Unless with extensive cross-validation, leaving only 15 out of 300 images for testing does not exactly allow for a robust evaluation. As the samples collected and annotated under controlled laboratory conditions are already bound to be somewhat similar to one another, the likelihood of overfitting is high, if the reported accuracy for each tested architecture only refers to that on the small test split. Was the same set of test images used for all architectures tested?

We agree that this is unusual, but we followed the guidelines published by DLC.

We evaluated the trained network as reported in the revised manuscript, but agree with the reviewer that it is important to also include information for potential users that in general pose estimation networks need to be thoroughly evaluated before use in online experiments. Additionally, to the sections already included in the revised manuscript concerning this issue, we added the following information (Page 14, Line 39-41):

“Note that for some cases a small number of test images (5%, 15) might require further evaluation of the trained network to guarantee sufficient accuracy and generalization.”

To directly answer the question of the reviewer: Yes, the same set of test images was used for all architectures with the same number of body parts as stated in the methods section of the revised manuscript (Page 14, line 44-48).

- P16, line 5-6: Please elaborate on whether one or multiple scorers annotated the human labelled dataset and if discrepancies between observers were quantified.

The human labelled dataset was collected by a single human annotator, now specified in the revised manuscript (Page 15, line 8).

The sentence now reads as follows:

„To evaluate our own DLC trained networks, we measured the pose estimation error and compared it to a human labeled dataset **(labeled by a single human annotator).**”

Reviewer #2 (Remarks to the Author):

I am very happy with the additional data and discussion included by the authors. I feel they have gone above and beyond in responding to my comments.

We again thank the reviewer for the time to thoroughly review our manuscript.

Reviewer #3 (Remarks to the Author):

The authors have done an excellent job of responding to reviewer comments. The two example use cases are much more clearly presented, and the documentation of hardware latencies, network inference framerates, and error relative to human annotations are all important additions. I do not believe that any further work on the paper contents is needed.

Thanks for thoroughly reviewing and improving our manuscript.